# Nucleoporin 107, 62 and 153 mediate *Kcnq1ot1* imprinted domain regulation in extraembryonic endoderm stem cells

Saqib S. Sachani [ID] [1,2,3,4], Lauren S. Landschoot[1,2], Liyue Zhang[1,2], Carlee R. White[1,2], William A. MacDonald[3,4], Michael C. Golding [ID] [5] & Mellissa R.W. Mann [ID] [3,4]

Genomic imprinting is a phenomenon that restricts transcription to predominantly one parental allele. How this transcriptional duality is regulated is poorly understood. Here we perform an RNA interference screen for epigenetic factors involved in paternal allelic silencing at the *Kcnq1ot1* imprinted domain in mouse extraembryonic endoderm stem cells. Multiple factors are identified, including nucleoporin 107 (NUP107). To determine NUP107's role and specificity in *Kcnq1ot1* imprinted domain regulation, we deplete *Nup107*, as well as *Nup62, Nup98/96* and *Nup153*. *Nup107, Nup62* and *Nup153*, but not *Nup98/96* depletion, reduce *Kcnq1ot1* noncoding RNA volume, displace the *Kcnq1ot1* domain from the nuclear periphery, reactivate a subset of normally silent paternal alleles in the domain, alter histone modifications with concomitant changes in KMT2A, EZH2 and EHMT2 occupancy, as well as reduce cohesin interactions at the *Kcnq1ot1* imprinting control region. Our results establish an important role for specific nucleoporins in mediating *Kcnq1ot1* imprinted domain regulation.

[1] Departments of Obstetrics & Gynaecology, and Biochemistry, Western University, Schulich School of Medicine and Dentistry, London, ON N6A 5W9, Canada. [2] Children's Health Research Institute, London, ON N6C 2V5, Canada. [3] Departments of Obstetrics, Gynecology and Reproductive Sciences, University of Pittsburgh School of Medicine, Pittsburgh, PA 15213, USA. [4] Magee-Womens Research Institute, Pittsburgh, PA 15213, USA. [5] Department of Veterinary Physiology, College of Veterinary Medicine, Texas A&M University, College Station, TX 77843, USA. These authors contributed equally: Saqib S. Sachani, Lauren S. Landschoot. Correspondence and requests for materials should be addressed to M.R.W.M. (email: mannmr@mwri.magee.edu)

Genomic imprinting is an epigenetic process that restricts expression of specific genes to predominantly the maternally- or paternally inherited allele. The biochemical mechanisms that generate this allele-specific asymmetry rely upon multiple protein families, broadly termed epigenetic factors. However, it is still not fully clear which specific epigenetic factors establish and maintain this duality. In this study, we investigated the *Kcnq1ot1* domain to further understand the mechanisms involved in allele-specific asymmetry. The *Kcnq1ot1* domain serves as an excellent model of imprinted domain regulation, since all known epigenetic regulatory mechanisms have some role at the *Kcnq1ot1* imprinted domain, including differential DNA methylation and chromatin modifications, noncoding RNA expression, transcriptional interference, noncoding RNA-mediated silencing, CCCTC-binding factor (CTCF)/cohesin insulator activity and chromatin looping[1–9].

Within the *Kcnq1ot1* domain resides the imprinting control region (ICR), the paternally expressed *Kcnq1ot1* (*Kcnq1* opposite transcript 1) noncoding RNA (ncRNA), 9 maternally expressed protein-coding genes, and several genes that escape imprinting[1,4,8,10,11]. On the maternal allele, the *Kcnq1ot1* ICR is methylated, silencing the embedded *Kcnq1ot1* promoter and its transcription, thereby permitting expression of neighboring genes. On the paternal allele, the *Kcnq1ot1* ICR is unmethylated, allowing *Kcnq1ot1* ncRNA transcription, which results in protein-coding gene silencing. In mice, paternal inheritance of a *Kcnq1ot1* ICR deletion leads to paternal reactivation of imprinted genes within the domain at midgestation[12]. Similarly, paternal transmission of *Kcnq1ot1* ncRNA truncations result in paternal allelic reactivation in midgestation embryos[5,6]. These results indicate that the *Kcnq1ot1* ICR as well as *Kcnq1ot1* ncRNA/transcription are essential for paternal allelic silencing.

To date, several epigenetic factors have been identified that regulate *Kcnq1ot1* domain imprinting, including polycomb repressive complex (PRC) 1 and 2 proteins (E3 ubiquitin-protein ligase RING2 (RNF2), enhancer of zeste homolog 2 (EZH2), embryonic ectoderm development (EED)), histone methyltransferase 2 (EHMT2/G9a), suppressor of variegation 4-20 homolog 1 (SUV420H1) and DNA methyltransferase 1 (DNMT1)[3,8,9,13–16].

Here, we identify multiple epigenetic factors involved in *Kcnq1ot1* imprinted domain regulation, including a nucleoporin-dependent mechanism. We show that NUP107, NUP62, and NUP153 are required in extraembryonic endoderm stem cells to maintain *Kcnq1ot1* ncRNA expression and volume at the domain, to position the *Kcnq1ot1* imprinted domain at the nuclear periphery, as well as to silence a subset of paternal alleles of the protein-coding genes in the domain. We also show that nucleoporins regulate imprinted gene expression through active and repressive histone modifications but not DNA methylation at the ICR. Lastly, we show nucleoporins direct the occupancy of cohesin complex proteins at the paternal *Kcnq1ot1* ICR.

## Results

**Multiple epigenetic factors silence a paternal *NeoR* cassette.** To identify epigenetic factors involved in paternally inherited *NeoR* silencing, as a proxy for paternal allelic silencing of imprinted genes in the *Kcnq1ot1* domain, a positive-selection, loss-of-function RNA interference screen was performed using a library of short hairpin RNAs (shRNAs) for 250 epigenetic factors, with ~3 hairpins per factor[17] (Supplementary Fig. 1). To conduct this screen, we used an existing transgenic mouse model, where exons 1 and 2 of the *Cdkn1c* imprinted gene were replaced with the PGK-neomycin resistance cassette (*Cdkn1c*[+/ΔneoR])[18] (Supplementary Fig. 2a). Our strategy was predicated upon silencing of

the paternally inherited drug resistance marker (*Cdkn1c*[ΔneoR])[19] in CAST7XB6 *Cdkn1c*[+/ΔneoR] embryonic, trophoblast and extraembryonic endoderm (XEN) stem cells. Reactivation of the silent *Cdkn1c*[ΔneoR] allele following depletion allowed for survival and selection of colonies in the presence of neomycin, and thus, identification of epigenetic factors crucial in maintaining its silent state. Albeit, only XEN cells displayed repression of the paternally inherited *Cdkn1c*[ΔneoR] allele to a level that would allow efficient screening (Supplementary Fig. 2b). Using this strategy, 696 colonies were picked for a second round of neomycin selection, following which 297 colonies were isolated. DNA was sequenced to identify shRNA-targeted factors controlling *Cdkn1c*[ΔneoR] repression (Supplementary Fig. 1). In total, 41 epigenetic modifiers were identified (Table 1), with stronger candidates having multiple independent colonies and hairpins recovered. Candidates included factors (RNF2, EZH2, EED, SUV420H1, and DNMT1) previously shown to play a role in *Kcnq1ot1* imprinted domain regulation[3,7,9,13–16,20], validating our screening strategy. However, they were recovered at a lower frequency than other candidates. Given the chance of off-target effects, validation of candidates will be required to delineate their role in paternal allelic silencing at the *Kcnq1ot1* domain. Here, we focused on the candidate nucleoporin 107 (*Nup107*) (Table 1). Nucleoporins are proteins that constitute nuclear pore complexes that stud the nuclear membrane, allowing nuclear-cytoplasmic transport[21]. A more recent role identified for nucleoporins is gene regulation, although the mechanistic underpinnings of this regulation are not fully understood[22–28].

**Nucleoporins regulate *Kcnq1ot1* ncRNA expression and volume.** To investigate the role and specificity of NUP107, we examined NUP107, as well as three additional nucleoporins with documented chromatin association[23,27,28] (NUP62, NUP98 and NUP153, not included in our original screen) for their role in regulating the *Kcnq1ot1* imprinted domain. B6XCAST XEN cells were transfected with two sets of *Nup107*, *Nup62*, *Nup98/96* and *Nup153* siRNAs to produce RNA and protein depletion (Supplementary Fig. 3a–d, antibody validation; Supplementary Fig. 4a, b). Cells were then assessed for total and allele-specific *Kcnq1ot1* ncRNA expression. Compared to controls, *Nup107*, *Nup62* and *Nup153* depletion significantly reduced *Kcnq1ot1* ncRNA levels to 0.25, 0.45 and 0.76 of controls, respectively, while *Nup98/96* depletion produced a 1.88 times increase in *Kcnq1ot1* ncRNA levels (Fig. 1a). For all *Nup* depletions, the *Kcnq1ot1* ncRNA maintained paternal expression (Fig. 1b), except for *Nup153* depletion, where the normally silent maternal *Kcnq1ot1* ncRNA was reactivated (24%). To determine absolute maternal and paternal expression levels instead of allelic ratios, we developed a highly sensitive, precision method for assessing absolute transcript abundance of each allele independently using droplet digital PCR with FAM and HEX strain-specific probes (Supplementary Fig. 5). Compared to control cells, there was a significant decrease in paternal *Kcnq1ot1* transcripts by 9928-11,116 copies upon *Nup107*, *Nup62*, and *Nup153* depletion, and a significant increase in paternal *Kcnq1ot1* transcripts by 18,805 copies upon *Nup98/96* depletion (Fig. 1c). While *Nup107*, *Nup62*, and *Nup98/96* depletion did not change the low number of maternal *Kcnq1ot1* transcripts, *Nup153* depletion produced a significant increase in maternal *Kcnq1ot1* transcripts by 4093 copies, compared to controls.

Paternal *Kcnq1ot1* ncRNA transcripts coat the paternal *Kcnq1ot1* domain in what is thought to be a repressive nuclear compartment[8,11,15,29]. To determine whether reduced *Kcnq1ot1* ncRNA abundance altered nuclear *Kcnq1ot1* ncRNA volume, 3D *Kcnq1ot1* RNA/DNA FISH was performed on G1-synchronized

**Table 1 Candidate epigenetic factors of the *Kcnq1ot1* imprinted domain**

| Candidate | Gene name | Total colonies | Number of shRNAs |
|---|---|---|---|
| Ezh1 | Enhancer of zeste 1 | 21 | 1 |
| Smarca5 | Smarcᵃ subfamily a, member 5 | 16 | 1 |
| Trdmt1 | tRNA aspartic acid methyltransferase 1 (Dnmt2) | 15 | 1 |
| Taf6l | TATA-box binding protein associated factor 6 like | 11 | 1 |
| Smarcc2 | Smarcᵃ subfamily c, member 2 | 9 | 4 |
| Smarce1 | Smarcᵃ, subfamily e, member 1 | 9 | 2 |
| Nup107 | Nucleoporin 107 | 9 | 1 |
| Atrx | Alpha thalassemia/mental retardation syndrome X-linked | 9 | 1 |
| Smarcad1 | Smarcᵃ subfamily a, containing DEAD/H box 1 | 8 | 3 |
| Kat2a | K(lysine) acetyltransferase 2A | 8 | 1 |
| Mbd4 | Methyl-CpG binding domain protein 4 | 7 | 1 |
| Kdm4a | Lysine (K)-specific demethylase 4A | 7 | 1 |
| Smarce1-ps1 | Smarcᵃ, subfamily e, member 1 pseudogene 1 | 6 | 2 |
| Kat7 | K(lysine) acetyltransferase 7 | 6 | 2 |
| Suv420h1 | Suppressor of variegation 4-20 homolog 1 | 6 | 1 |
| Mbd3 | Methyl-CpG binding domain protein 3 | 6 | 1 |
| Hdac9 | Histone deacetylase 9 | 5 | 2 |
| Hdac1 | Histone deacetylase 1 | 5 | 1 |
| Sin3a | Transcriptional regulator, SIN3A | 4 | 2 |
| Dnmt3a | DNA methyltransferase 3A | 4 | 2 |
| Smyd3 | SET and MYND domain containing 3 | 4 | 1 |
| Sirt1 | Sirtuin 1 | 3 | 3 |
| Rnf2 | Ring finger protein 2 | 3 | 3 |
| Ezh2 | Enhancer of zeste 2 | 3 | 2 |
| Pcgf6 | Polycomb group ring finger 6 | 3 | 1 |
| Kat6b | K(lysine) acetyltransferase 6B | 3 | 1 |
| Snx10 | Sorting nexin 10 | 3 | 1 |
| Dnmt1 | DNA methyltransferase 1 | 2 | 2 |
| Hdac2 | Histone deacetylase 2 | 2 | 2 |
| Smarcd2 | Smarcᵃ subfamily d, member 2 | 2 | 2 |
| Ep300 | E1A binding protein p300 | 2 | 1 |
| Hltf | Helicase-like transcription factor (Smarca3) | 2 | 1 |
| Hdac8 | Histone deacetylase 8 | 2 | 1 |
| Eed | Embryonic ectoderm development | 2 | 1 |
| Taf5l | TATA-box binding protein associated factor 5 like | 1 | 1 |
| Phc3 | Polyhomeotic-like 3 | 1 | 1 |
| Kat2b | K(lysine) acetyltransferase 2B | 1 | 1 |
| Kmt2d | Lysine (K)-specific methyltransferase 2D | 1 | 1 |
| Hdac5 | Histone deacetylase 5 | 1 | 1 |
| Hdac7 | Histone deacetylase 7 | 1 | 1 |
| Smarca1 | Smarcᵃ subfamily a, member 1 | 1 | 1 |

ᵃSmarc, SWI/SNF-related, matrix associated actin-dependent regulator of chromatin

control and *Nup*-depleted XEN cells, and *Kcnq1ot1* ncRNA volume was calculated. *Kcnq1ot1* ncRNA signal was restricted to the paternal *Kcnq1ot1* domain in controls, with a reduction in cells with paternal *Kcnq1ot1* ncRNA signal in *Nup107*-, *Nup62*-, and *Nup153*- but not *Nup98/96*-depleted XEN cells (Fig. 2a, b). *Nup153*-depleted XEN cells also displayed *Kcnq1ot1* ncRNA signal at the maternal domain in 32% of cells. With respect to *Kcnq1ot1* ncRNA volume, the majority of control cells possessed medium volumes (70–72%; low, 13–18%; high, 11–15%; very high, 0–1%) (Fig. 2d). Consistent with altered *Kcnq1ot1* ncRNA levels, *Nup107*, *Nup62*, and *Nup153* depletion generated a significant increase in cells with low *Kcnq1ot1* ncRNA volumes (83, 76, 26%, respectively), while *Nup98/96* depletion significantly increased the number of cells with high (35%) or very high (15%) volumes (Fig. 2d). In addition, upon *Nup153* depletion, 76% of cells with maternal *Kcnq1ot1* reactivation had low *Kcnq1ot1* ncRNA volumes. Notably, changes in *Kcnq1ot1* ncRNA abundance and volume were not due to altered *Kcnq1ot1* transcript stability (Supplementary Fig. 6). Together, these results demonstrate that nucleoporins facilitate paternal *Kcnq1ot1* ncRNA expression.

**NUPs position the *Kcnq1ot1* domain to the nuclear periphery.** Previous studies demonstrated that the *Kcnq1ot1* ncRNA-coated domain can be situated at the nuclear periphery[7,8], and that the distance between distal probes within the *Kcnq1ot1* domain differed for maternal and paternal alleles[11,15]. To examine the position of the *Kcnq1ot1* domain in XEN cells, distance of the *Kcnq1ot1* DNA signal centroid from the nuclear rim was calculated[30,31]. Here, size of the *Kcnq1ot1* DNA volume identified the maternal ($0.1–0.8\ \mu m^3$) and paternal ($0.9–1.3\ \mu m^3$) domains (Supplementary Fig. 7). In control cells, the paternal *Kcnq1ot1* domain was stationed at the nuclear periphery, subnuclear periphery and nuclear interior in 92, 4–6, and 3–4% of cells, respectively (Fig. 2e). In *Nup107*-, *Nup62*- and *Nup153*-depleted XEN cells, nuclear periphery positioning of the paternal *Kcnq1ot1* domain was significantly reduced to 54, 61, and 64% of the cells, shifting to the subnuclear periphery (26, 33, and 17%) and nuclear interior (20, 6, and 19%), respectively. Furthermore, maternal *Kcnq1ot1* domain positioning was significantly increased at the nuclear periphery in *Nup153*-depleted XEN cells possessing a maternal *Kcnq1ot1* ncRNA signal (+; 60%) compared to control cells (14–17%) and *Nup153*-depleted XEN cells without a maternal *Kcnq1ot1* ncRNA signal (−; 9%). *Nup98/96*

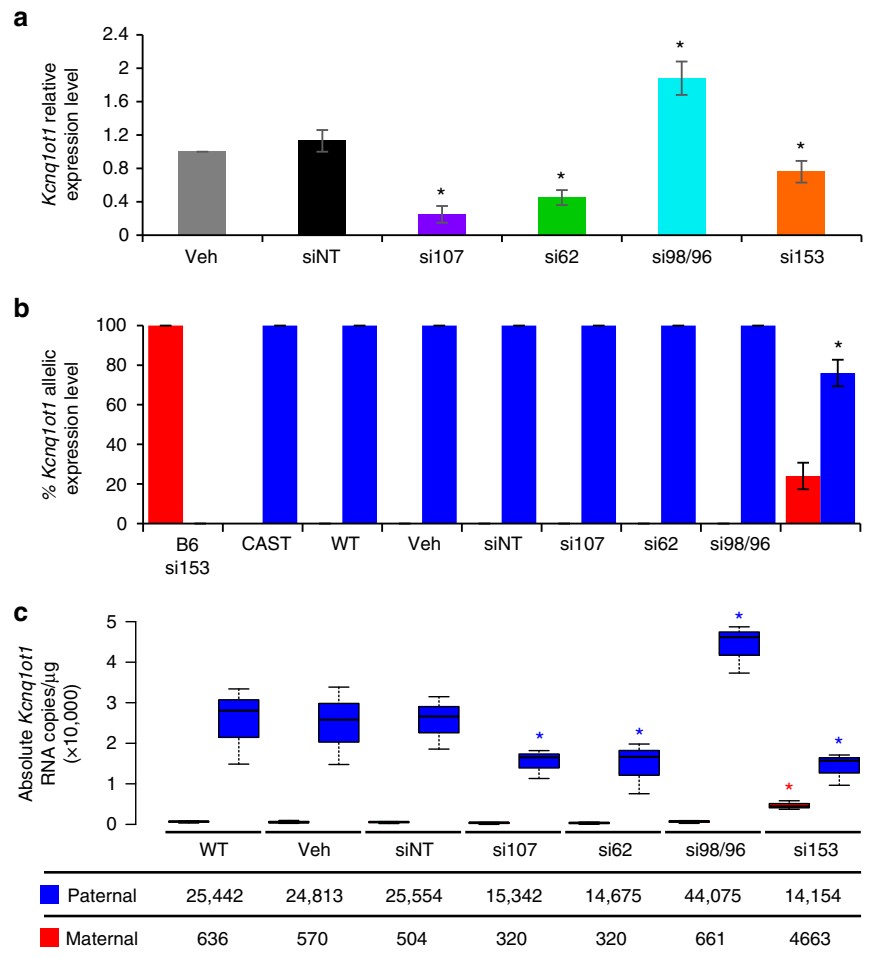

**Fig. 1** Nucleoporin depletion disrupted *Kcnq1ot1* ncRNA expression. **a** Real-time *Kcnq1ot*1 ncRNA expression levels normalized to *Gapdh* ($n = 3$ biological samples with four technical replicates per sample). **b** Allelic *Kcnq1ot*1 ncRNA expression in control and *Nup*-depleted XEN cells ($n = 3$ biological samples; $n = 4$ technical replicates per sample). **c** Absolute allelic *Kcnq1ot*1 transcript abundance determined by droplet digital PCR in control and *Nup*-depleted XEN cells, as measured by RNA copies μg$^{-1}$ ($n = 3$ biological samples). Center lines, medians; box limits, 25th and 75th percentiles as determined by R software; whiskers, 1.5 times the interquartile range from 25th and 75th percentiles; B6/maternal, red; CAST/paternal, blue; error bars, s.e.m.; *, significance $p < 0.05$ compared to siNT control (*t* test); WT wild type, Veh vehicle, siNT nontargeting siRNA, si107 *Nup107* siRNA, si62 *Nup62* siRNA, si98/96 *Nup98/96* siRNA, si153 *Nup153* siRNA

depletion also significantly reduced paternal *Kcnq1ot1* domain nuclear peripheral (84%) positioning, albeit to a lesser extent than other *Nup* depletions. Next, we examined the relationship between *Kcnq1ot1* ncRNA volume and nuclear positioning. The majority of control cells had medium paternal *Kcnq1ot1* ncRNA volume that resided at the nuclear periphery (Supplementary Fig. 8). By comparison, in *Nup107*- and *Nup62*-depleted cells with low paternal *Kcnq1ot1* ncRNA volume, the *Kcnq1ot1* domain was displaced from the nuclear periphery. In *Nup153*-depleted cells, paternal *Kcnq1ot1* domains with low and medium *Kcnq1ot1* ncRNA volumes shifted away from the nuclear periphery, while maternal domains with low and medium *Kcnq1ot1* ncRNA volumes shifted toward the nuclear periphery. In contrast, *Nup98/96*-depleted cells with increased paternal *Kcnq1ot1* ncRNA volumes primarily retained nuclear peripheral positioning of the *Kcnq1ot1* domain. These results indicate that NUP107, NUP62, and NUP153 target the *Kcnq1ot1* ncRNA-coated domain to the nuclear periphery.

**NUPs bind the *Kcnq1ot1* ICR and imprinted gene promoters.** We next investigated nucleoporin interactions with the *Kcnq1ot1* imprinted domain. Quantitative chromatin immunoprecipitation

(ChIP) was performed using mAb414 (primarily interacts with NUP62, NUP107, and NUP160 in XEN cells (Supplementary Fig. 3b)) and NUP153 antibodies; ChIP-grade NUP107 and NUP62 antibodies were not available. Antibodies were first validated at predictive positive and negative sites identified from mouse ES cells NUP153 DNA adenine methyltransferase identification sequencing (DamID-seq) data[27] (Supplementary Fig. 9). To investigate nucleoporin interactions at the *Kcnq1ot1* domain, 21 sites across the *Kcnq1ot1* ICR through to the reported H3K4me1-enriched enhancer element[32,33], 1–2 sites at imprinted gene promoters, and two control regions were assessed for total and allele-specific binding in WT XEN cells (Supplementary Fig. 10a, Fig. 3a). Significant mAb414 (NUP107/62) enrichment was observed at two sites within the *Kcnq1ot1* ICR (IC3, 100 bp upstream; IC4, 1.7 kb downstream of the *Kcnq1ot1* transcription start site), the *Kcnq1ot1* enhancer element (E1 and E2) and the *Osbpl5* promoter (Os1 and Os2) on the paternal allele (Supplementary Fig. 10b, Fig. 3b, d). Significant NUP153 enrichment was observed at the *Kcnq1ot1* ICR (IC3, IC4), where both parental alleles were equally enriched, and at the paternal *Kcnq1* (Kc1, Kc2) and *Cd81* (Cd1) promoters (Supplementary Fig. 10b, Fig. 3c, e). By comparison, no NUP98 enrichment was observed at the

*Kcnq1ot1* ICR or enhancer element (Supplementary Fig. 10b). Next, we investigated whether nucleoporin−chromatin interactions were lost upon *Nup* depletion. Since the mAb414 antibody recognizes NUP62 and NUP107 (Supplementary Fig. 3b), double depletion was performed. We found a significant decrease in nucleoporin occupancy at the *Kcnq1ot1* ICR and enhancer element in double-depleted cells at the paternal allele (Fig. 3d). Upon *Nup153* depletion, NUP153 binding on the paternal and maternal *Kcnq1ot1* ICR, and the paternal *Kcnq1* and *Cd81* promoters was significantly reduced (Fig. 3e). To determine whether there was interdependency between NUP107/62 and NUP153 binding, *Nup153*-depleted cells were examined for mAb414

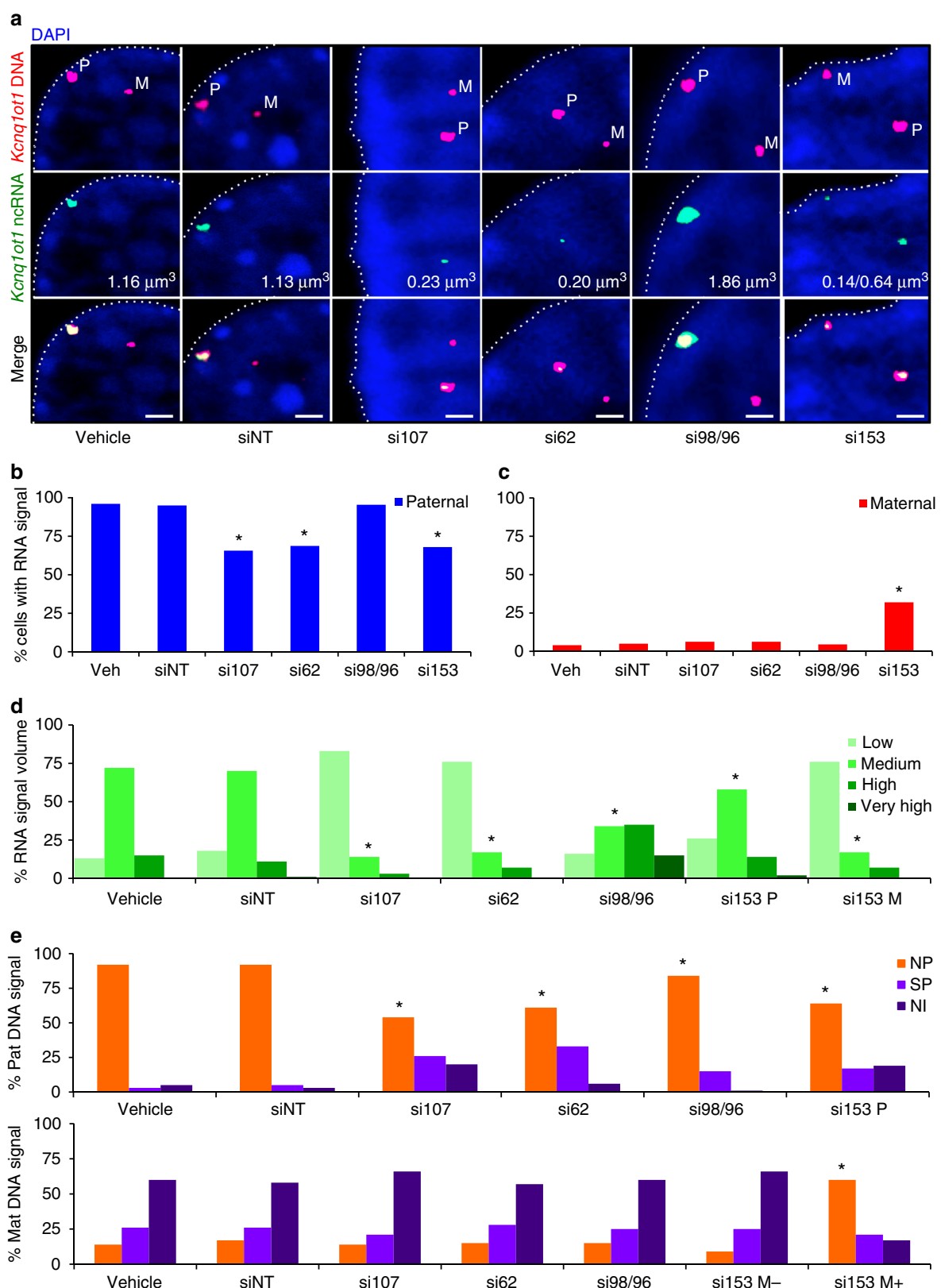

enrichment, and *Nup107*- and *Nup62*-depleted cells were assessed for NUP153 enrichment. *Nup153* depletion significantly decreased mAb414 (NUP107/62) binding at the *Kcnq1ot1* ICR region but not at the *Osbpl5* promoter and enhancer sites (Fig. 3f). Conversely, *Nup107* and *Nup62* depletion significantly decreased NUP153 occupancy at the *Kcnq1ot1* ICR and *Kcnq1* promoter, while no change in enrichment was observed at the *Cd81* promoter (Fig. 3g). These results indicate that nucleoporin interactions at the paternal *Kcnq1ot1* ICR are cooperatively mediated through NUP107/NUP62 and NUP153.

We further investigated whether nucleoporins can interact with a *Kcnq1ot1* ICR DNA fragment in nuclear lysates from control and *Nup*-depleted XEN cells. Electromobility shift assays were first performed using biotin-labeled *Vim*, *Shank2* and *Dchr7* promoter fragments as controls. For the *Kcnq1ot1*, supershifts were observed for the biotin-labeled *Kcnq1ot1* ICR fragment with mAb414 (NUP107/62), NUP107, and NUP153 antibodies but not for the biotin-labeled, 2-kb upstream control fragment (Supplementary Fig. 11a, b), pointing to a direct interaction of these proteins with the *Kcnq1ot1* ICR. No supershift was produced using the NUP98 antibody. Next, reverse ChIP was performed to isolate proteins bound to a biotin-labeled *Kcnq1ot1* ICR and negative control fragments, followed by western blot analysis using NUP107, mAb414, NUP98, and NUP153 antibodies. NUP107 bound to the *Kcnq1ot1* ICR fragment but not the 2-kb upstream control fragment (Supplementary Fig. 12). *Nup107*-depleted lysates abrogated NUP107 binding, as well as NUP62 and NUP153 binding, indicating that NUP62 and NUP153 binding at the *Kcnq1ot1* ICR required NUP107 binding. NUP153 also bound directly to the *Kcnq1ot1* ICR fragment but not the control fragment. Upon *Nup153* depletion, binding to the *Kcnq1ot1* ICR fragment was lost, although NUP107 and NUP62 binding was not altered, perhaps indicating sequential binding of NUP107, NUP62, and NUP153.

**NUPs regulate paternal allelic silencing of imprinted genes.** Since NUP107, NUP62, and NUP153 were bound within the *Kcnq1ot1* domain, allelic expression of imprinted genes in the *Kcnq1ot1* domain was assessed by a conventional method of allelic ratio quantification[1]. *Nup107*, *Nup62*, and *Nup153* depletion resulted in paternal reactivation of the core group of genes, *Slc22a18*, *Cdkn1c*, and *Kcnq1* (Supplementary Fig. 13), as well as gene-specific reactivation (paternal *Osbpl5* and *Phlda2* alleles upon *Nup107* depletion, the paternal *Osbpl5* allele upon *Nup62* depletion, and the paternal *Cd81* allele upon *Nup62* depletion). By comparison, *Nup98/96* depletion had no effect on paternal allelic expression. To determine the absolute transcript abundance of maternal and paternal RNA copies, we developed allelic droplet digital PCR assays using FAM and HEX strain-specific probes for imprinted genes in the domains (Fig. 4, Supplementary Fig. 14). Compared to controls, we observed that *Nup107* and *Nup62* depletion increased paternal *Osbpl5* transcripts by 13,472–13,662

copies, *Nup107* depletion increased *Phlda2* transcripts by 430 copies, *Nup107*, *Nup62*, and *Nup153* depletion increased paternal transcripts of *Slc22a18* by 29,547–32,214, *Cdkn1c* by 10,882–13,153, and *Kcnq1* by 728–805 copies, and *Nup153* depletion increased *Cd81* transcripts by 13,033 copies (Fig. 4, Supplementary Fig. 15). Notably, even though the normally silent maternal *Kcnq1ot1* allele was reactivated in a proportion of *Nup153*-depleted XEN cells, no change was observed in maternal transcript copies of imprinted genes upon *Nup153* depletion. Overall, our data indicate that paternal allelic reactivation was not domain-wide. Instead, NUP107, NUP62, and NUP153 regulate paternal allelic silencing of a core group of genes, *Slc22a18*, *Cdkn1c*, and *Kcnq1*, while paternal allele silencing of genes more distal to the *Kcnq1ot1* ICR, *Phlda2*, *Osbpl5,* and *Cd81*, was nucleoporin-specific.

**Nuclear-cytoplasmic transport not impaired by *Nup* depletion.** An alternate explanation for altered regulation of the *Kcnq1ot1* domain upon nucleoporin depletion is impaired nuclear-cytoplasmic transport. The nuclear pore complex is composed of multiple copies of ~30 nucleoporins (~500 proteins in total)[34]. Impaired function may result from incorrect assembly of the nuclear pore complex when one component is depleted. Examining the levels of various nucleoporins within different structural components of the nuclear pore (Supplementary Fig. 16a), we found no change in nucleoporin levels in *Nup*-depleted XEN cells compared to controls (Supplementary Fig. 16b), suggesting that nuclear pore complex assembly was not affected upon *Nup* depletion. Previous studies have shown that import of cargo containing a classical bipartite nuclear localization signal (NLS) was impaired in *NUP153*-depleted HeLa cells[35]. We found no change in protein import in control and *Nup*-depleted XEN cells as measured by levels of the transcription factor 3 (E47)-red fluorescence protein (RFP)-NLS construct[36] compared to ivermectin-treated cells, where RFP import was inhibited (Supplementary Fig. 17a), or between endogenous inner centromeric protein (INCENP) levels (Supplementary Fig. 17b) or LAMINB1 protein localization (Supplementary Fig. 17c). With respect to export, previous studies found aberrant nuclear mRNA export in *NUP107*-depleted HeLa cells, with abnormal accumulation of polyA-mRNA in depleted nuclei[37]. Here, no significant difference in nuclear polyA-mRNA retention levels was observed between control and *Nup*-depleted XEN cells (Supplementary Fig. 17d). With regard to passive bidirectional diffusion, we found that *Nup*-depleted cells exhibited comparable levels of nuclear and cytoplasmic GFP mRNA as well as nuclear GFP protein from a *Luciferase* gene tagged with a GFP reporter protein (shLucGFP)[17,38,39] (Supplementary Fig. 17e). Since aberrant nuclear transport may be expected to compromise cellular function[37], a final test was conducted to measure XEN cell growth rate. No significant change in XEN cell growth rate or doubling time was observed in control and *Nup*-depleted XEN cells in a

**Fig. 2** Nucleoporin depletion disrupted *Kcnq1ot1* ncRNA volume and *Kcnq1ot1* domain positioning at the nuclear periphery. **a** Representative confocal nuclear images displaying *Kcnq1ot1* DNA (magenta) *Kcnq1ot1* ncRNA (cyan) and DAPI staining (blue) for G1-synchronized control and *Nup*-depleted XEN cells ($n = 4$; cell number = 109–123); upper panel, DNA FISH; middle panel, RNA FISH; lower panel, merge; M maternal domain, P paternal domain; white dashed line denotes nuclear rim. In these images, red and green fluorescence was converted to magenta and cyan. **b**, **c** Percent of cells with paternal or maternal *Kcnq1ot1* ncRNA signals. **d** Percent of cells with *Kcnq1ot1* ncRNA signal volume; low, 0–0.7 μm³; medium, 0.7–1.4 μm³; high, 1.4–2.1 μm³; very high, >2.1 μm³. **e** Distance of the paternal and maternal *Kcnq1ot1* domain from the nuclear membrane in control and *Nup*-depleted XEN cells. The maternal *Kcnq1ot1* domain was randomly positioned within the nucleus (expected nuclear periphery (NP) 15%; subnuclear periphery (SP) 30%; nuclear interior (NI) 60%), except for *Nup153*-depleted cells with a *Kcnq1ot1* ncRNA signal (si153 M+). For these analyses, cells with no RNA but detectable DNA FISH signals were included, while those lacking DNA signals were excluded. NP, 0–0.5 μm; SP, 0.5–1.5 μm; NI, 1.5–4 μm; error bars, s.e.m.; *, significance $p < 0.05$ compared to vehicle control (*t* test); WT wild type, Veh vehicle, siNT nontargeting siRNA, si107 *Nup107* siRNA, si62 *Nup62* siRNA, si98/96 *Nup98/96* siRNA, si153 *Nup153* siRNA, si153 M- *Nup153*-depleted cells without a *Kcnq1ot1* ncRNA signal; scale bar, 1 μm; $n = 4$; cell count number = 109–123

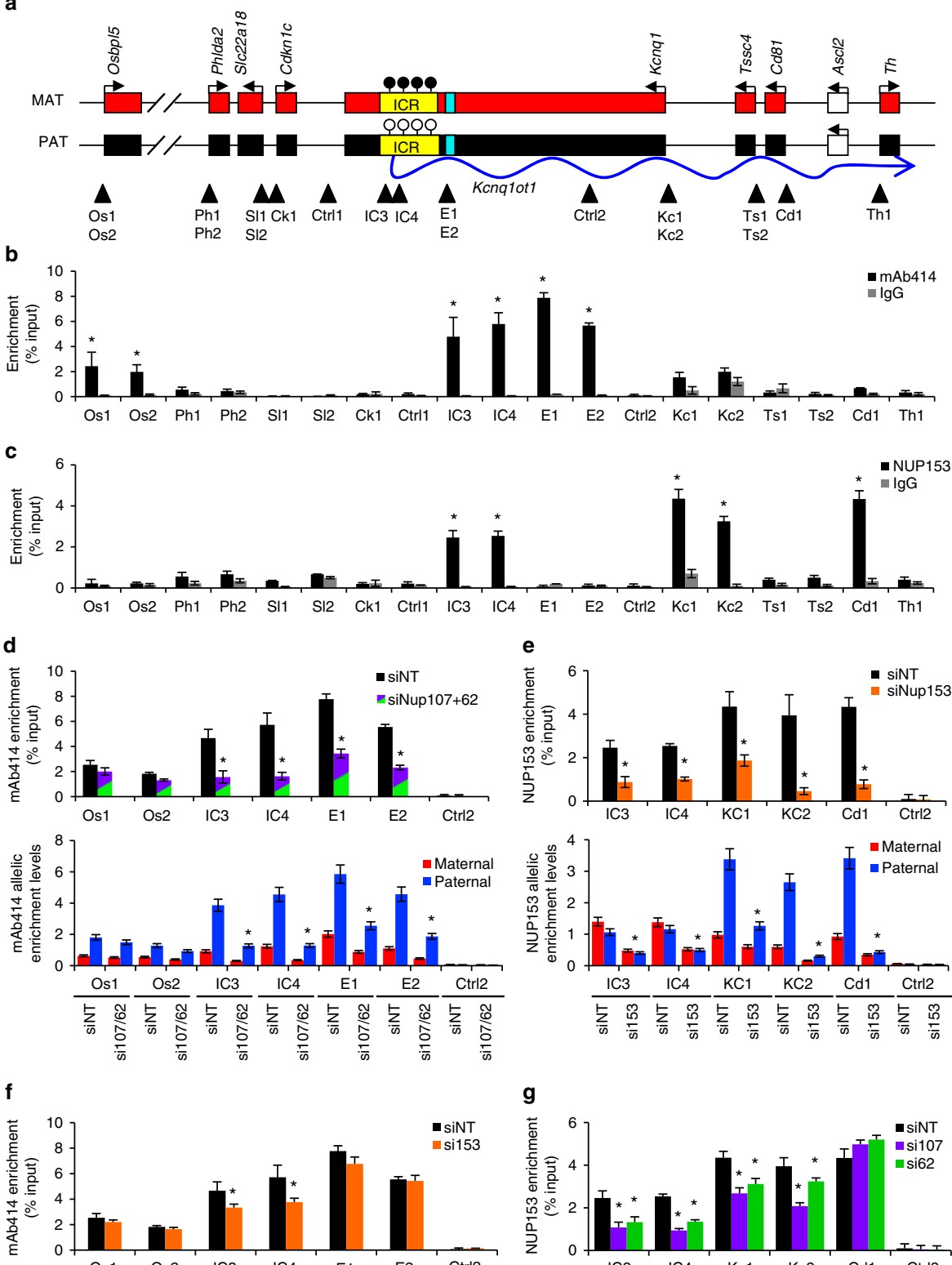

**Fig. 3** NUP107/62 and NUP153 interaction with the *Kcnq1ot1* ICR, and the *Osbpl5, Kcnq1, Cd81* promoters. **a** The *Kcnq1ot1* domain with regions of analysis (arrowheads) at the *Kcnq1ot1* ICR, IC3, IC4; enhancer element, E1, E2; imprinted gene promoters; Os1, Os2, Ph1, Ph2, Sl1, Sl2, Ck1, Kc1, Kc2, Ts1, Ts2, Cd1, Th1; and negative control sites, Ctrl1, Ctrl2. Quantitative ChIP analysis using **b** mAb414 antibodies and **c** NUP153 antibodies in wild-type XEN cells at regions across the domain, respectively ($n = 3$ biological samples with four technical replicates per sample). Quantitative allelic analysis for **d** mAb414 and **e** NUP153 in siNT- and nucleoporin-depleted XEN cells. Allelic proportions are represented as percent of the total enrichment levels ($n = 3$ biological samples with four technical replicates per sample). **f** Quantitative ChIP analysis using mAb414 antibodies was performed in control and *Nup153*-depleted XEN cells at sites of NUP153 enrichment ($n = 3$ biological samples with three technical replicates per sample). **g** Quantitative ChIP analysis using NUP153 antibodies was performed in control and *Nup107*- and *Nup62*-depleted cells at sites of mAb414 (NUP107/62) enrichment ($n = 3$ biological samples with three technical replicates per sample). Error bars, s.e.m.; *, significance $p < 0.05$ compared to the IgG or siNT control (*t* test)

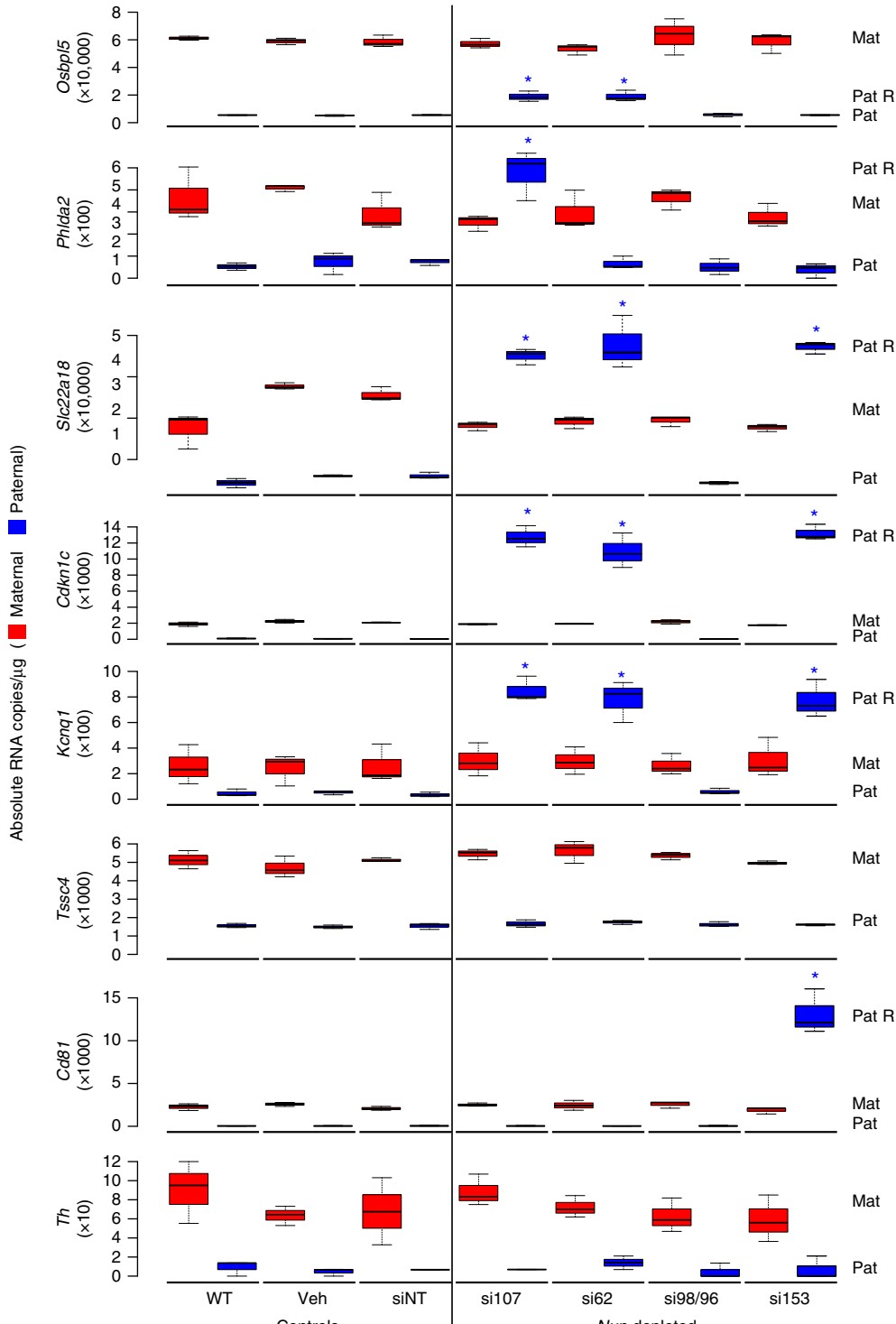

**Fig. 4** *Nup107, Nup62,* and *Nup153* depletion reactivate a subset of paternal alleles at the *Kcnq1ot1* domain. Absolute allelic transcript abundance of imprinted genes determined by droplet digital PCR in control and *Nup*-depleted XEN cells, as a measure of RNA copies µg$^{-1}$ ($n = 3$ biological samples). Center lines, medians; box limits, 25th and 75th percentiles as determined by R software; whiskers, 1.5 times the interquartile range from 25th and 75th percentiles; Mat, maternal; Pat, paternal; Pat R, reactivated paternally silent allele; *, significance $p < 0.05$ compared to the siNT control ($t$ test). Error bars, s.e.m.

direct cell counting assay (Supplementary Fig. 18a, b), which was similar to *NUP107*-depleted HeLa cells[37]. Overall, these results indicate that nuclear-cytoplasmic transport in XEN cells was not affected by nucleoporin depletion.

**DNA methylation not altered by *Nup* depletion.** Another explanation for a reduction in paternal *Kcnq1ot1* ncRNA levels

and subsequent paternal allelic reactivation upon *Nup107, Nup62,* and *Nup153* depletion could be a gain of DNA methylation at the normal unmethylated paternal ICR. In addition, reactivation of maternal *Kcnq1ot1* in *Nup153*-depleted XEN cells could be explained by a loss of DNA methylation at the methylated maternal ICR. Analysis of parental-specific methylation demonstrated that the maternal *Kcnq1ot1* ICR was hypermethylated

while the paternal ICR was hypomethylated in control and *Nup*-depleted XEN cells (Supplementary Fig. 19). Thus, changes in NUP107, NUP62, and NUP153 regulation at the *Kcnq1ot1* imprinted domain were not through alterations in *Kcnq1ot1* ICR methylation.

**NUPs maintain chromatin at *Kcnq1ot1* ICR and gene promoters.** We next investigated whether altered *Kcnq1ot1* ncRNA expression and paternal allelic silencing at the *Kcnq1ot1* imprinted domain is due to alterations in chromatin state[4,9,15,40]. ChIP assays were performed on control and depleted XEN cells using antibodies directed against RNAPII and histone 3 lysine 4 trimethylation (H3K4me3) as marks for active chromatin, and H3K9me2 and H3K27me3 for repressed chromatin. H3K4me3 and H3K27me3 peaks in XEN cells were used for primer placement[41]. Antibodies were first validated at the XEN cell expressed *FoxA2* gene and XEN cell repressed *MyoD* gene[17,42], where active and repressive modifications were observed, respectively (Supplementary Fig. 20). At the *Kcnq1ot1* domain, we first verified equal allelic detection in input chromatin (Supplementary Fig. 21) as well as equivalent enrichment for histone 3 (H3), a protein expected on both parental alleles (Supplementary Fig. 22). For the *Kcnq1ot1* ICR, there was a significant decrease in RNAPII and H3K4me3 enrichment and a significant increase in H3K9me2 and/or H3K27me3 enrichment on the paternal *Kcnq1ot1* ICR in *Nup107*-, *Nup62*-, and *Nup153*-depleted cells compared to control cells (Fig. 5a, b), which would account for reduced paternal *Kcnq1ot1* ncRNA levels. In addition, on the maternal *Kcnq1ot1* ICR, we observed a significant increase in H3K4me3 and RNAPII enrichment along with decreased repressive histone modifications upon *Nup153* depletion, which could account for reactivation of the maternal transcript. At the promoters of imprinted genes in the domain, there was significantly increased enrichment of RNAPII and/or H3K4me3 at the paternal *Slc22a18*, *Cdkn1c*, and *Kcnq1* alleles upon *Nup107*, *Nup62*, and *Nup153* depletion, at the paternal *Osbpl5* and *Phlda2* promoters upon *Nup107* and *Nup62* depletion, and at the paternal *Cd81* promoter upon *Nup153* depletion compared to control cells (Fig. 5a, b). This corresponded with significantly reduced levels of H3K9me2 and/or H3K27me3, thereby accounting for their paternal reactivation. No significant changes in histone modifications were observed upon *Nup98/96* depletion (Fig. 5a, b). These results demonstrate that NUP107, NUP62, and NUP153 act to regulate histone modifications at the *Kcnq1ot1* ICR and at specific imprinted gene promoters.

Changes in histone modifications could be a consequence of altered histone methyltransferases or RNAPII binding at the *Kcnq1ot1* ICR and imprinted gene promoters upon nucleoporin depletion. First, we ruled out that nucleoporin depletion per se did not alter histone 3 lysine 4 methyltransferase 2a (KMT2A), EHMT2, EZH2, and RNAPII protein levels in control and *Nup*-depleted XEN cells (Supplementary Fig. 23a, b). We found decreased KMT2A, and increased EHMT2 and EZH2 enrichment levels upon *Nup107/62* and *Nup153* depletion compared to control and *Nup98/96*-depleted cells (Fig. 6a–c). At imprinted gene promoters with altered histone modifications, KMT2a enrichment was increased on the paternal *Osbpl5*, *Phlda2*, *Slc22a18*, *Cdkn1c*, and *Kcnq1* alleles upon *Nup107/62* depletion, and on the paternal *Slc22a18*, *Cdkn1c*, *Kcnq1*, and *Cd81* alleles upon *Nup153* depletion, while a corresponding decrease in EHMT2 and EZH2 recruitment was found on the paternal *Osbpl5*, *Cdkn1c*, *Kcnq1*, and *Cd81* alleles upon *Nup107/62* and/or *Nup153* depletion. Surprisingly, no change in EHMT2 and EZH2 levels were observed at the *Slc22a18* promoter. Overall, these results suggest that changes in H3K4me3, H3K9me2, and

H3K27me3 levels were conferred by alteration in KMT2A, EHMT2, and EZH2 binding, respectively, at the *Kcnq1ot1* ICR and imprinted gene promoters upon *Nup107/62* and/or *Nup153* depletion.

**NUPs promote cohesin interactions at paternal *Kcnq1ot1* ICR.** Another mechanism that may alter *Kcnq1ot1* imprinted domain regulation is CTCF and/or cohesin complex binding. Previous studies identified two CTCF-binding sites within the *Kcnq1ot1* ICR that were bound by CTCF and the cohesin complex in mouse embryonic fibroblasts (MEFs)[2,43,44]. Thus, we investigate the relationship between CTCF, cohesin proteins, and nucleoporins at the *Kcnq1ot1* domain. Next, we ruled out that nucleoporin depletion per se altered CTCF, SMC1A, or SMC3 levels in control and *Nup*-depleted XEN cells (Supplementary Fig. 24a, b). As a positive control, CTCF, structural maintenance of chromosomes 1a (SMC1A), and 3 (SMC3) antibodies were validated in embryonic stem cells at the *H19* ICR[2], where strong CTCF, SMC1A, and SMC3 enrichment was observed on the maternal *H19* ICR (Supplementary Fig. 25). To determine whether CTCF, SMC1A, and SMC3 localized at the *Nup*-positive enrichment sites at the *Kcnq1ot1* domain, ChIP was performed using CTCF, SMC1A, and SMC3 antibodies in XEN cells. No CTCF occupancy was observed at the *Kcnq1ot1* ICR and enhancer region in XEN cells (Supplementary Fig. 25b). By comparison, SMC1A and SMC3 binding was significantly enriched at IC3 and IC4, but not at other tested sites, with preferentially cohesin binding at the paternal alleles (Fig. 7a, b). This binding was significantly decreased upon *Nup107*/*Nup62* and *Nup153* depletion, but not *Nup98*/96 depletion (Fig. 7a, b).

To determine whether nucleoporin depletion resulted in decreased occupancy at other sites outside the *Kcnq1ot1* domain, we tested SMC1A and SMC3 binding at positive and negative sites (ES cells ChIP sequencing data[45]) within the promoters of *Vim* (NUP153 positive), *Orai2* (NUP153, mAb414 and NUP98 positive), *Shank2* (NUP153, mAb414 and NUP98 positive), as well as the *Dhcr7* promoter (NUP153, mAb414, NUP98 negative site), in control and *Nup*-depleted cells. SMC1A and SMC3 recruitment was not altered at these sites upon nucleoporin depletion (Supplementary Fig. 26), indicating that SMC1A and SMC3 binding at *Vim*, *Orai2*, *Shank2*, and *Dhcr7* were nucleoporin-independent, whereas binding at the *Kcnq1ot1* ICR was *Nup107*-, *Nup62*-, and *Nup153*-dependent.

Finally, to gain an understanding of specific interactions between NUP107, NUP62, NUP153, and CTCF, SMC1A and SMC3, we performed immunoprecipitation assays using control and *Nup*-depleted nuclear lysates and mAb414 and NUP153 antibodies, followed by western analysis with CTCF, SMC1A, and SMC3 antibodies. We first established that NUP107/62 and NUP153 interacted with CTCF, SMC1A, SMC3 in control cells, and that these interactions were reduced in *Nup107/62*- and *Nup153*-depleted cells (Supplementary Fig. 27). We next determined whether interaction with CTCF, SMC1A, and SMC3 occurred selectively through NUP107/62, and/or NUP153. In *Nup107/62*-depleted cells, interactions between NUP153 and CTCF, SMC1A, and SMC3 were reduced compared to control cells (Fig. 8a). By comparison, in *Nup153*-depleted cells, mAb414 (NUP107/62) maintained CTCF, SMC1A, SMC3 interactions similar to controls (Fig. 8a), indicating that NUP107/62 was required for NUP153 interactions with CTCF, SMC1A, and SMC3. To delineate whether these interactions were mediated via NUP107 or NUP62, NUP153 interactions were investigated in *Nup107*- and *Nup62*-depleted cells. Notably, NUP107 but not NUP62 reduced NUP153 interactions with

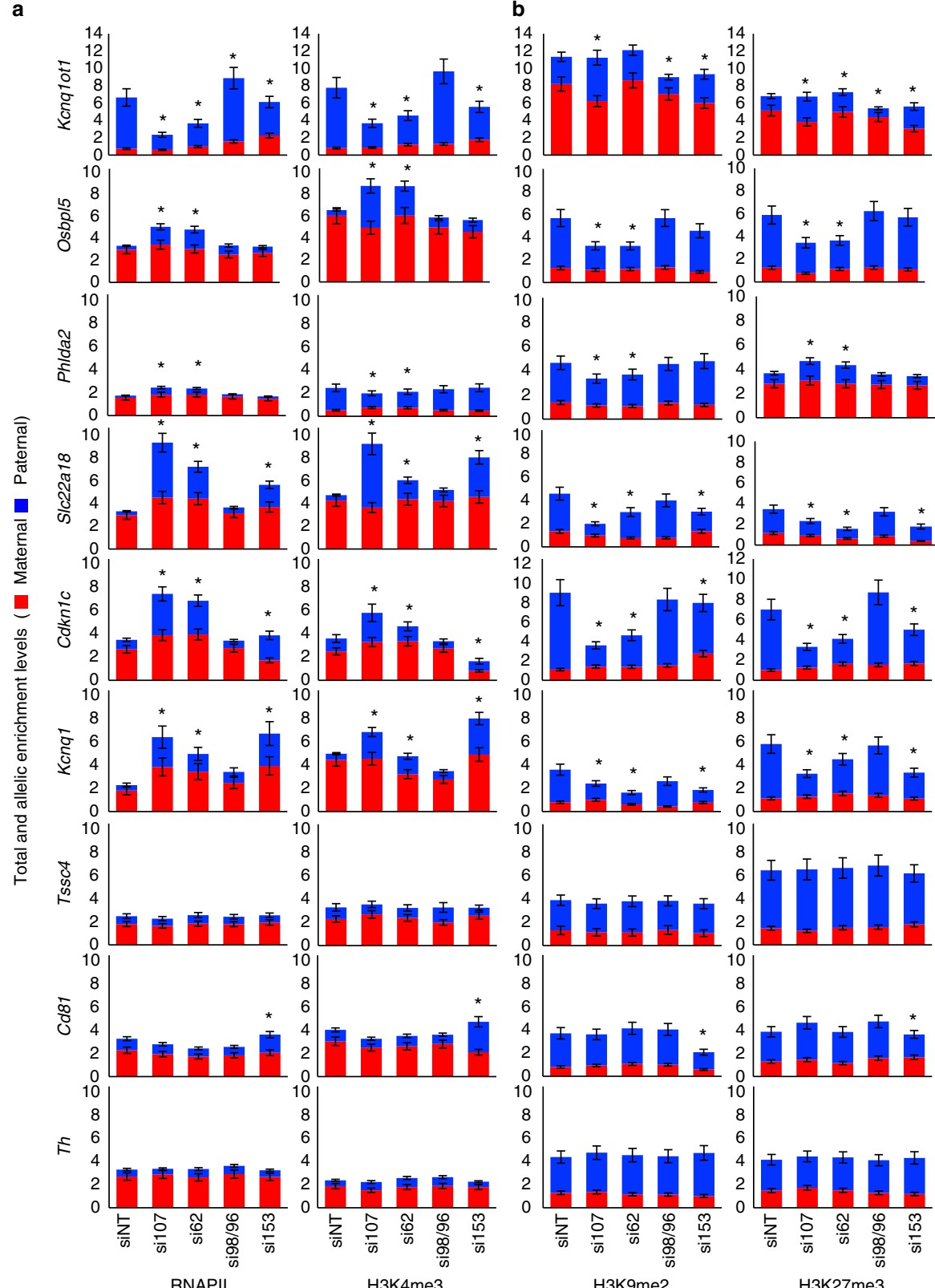

**Fig. 5** *Nup107, Nup62,* and *Nup153* depletion disrupts RNAPII enrichment and histone modifications at the *Kcnq1ot1* ICR and imprinted gene promoters. **a** RNAPII and H3K4me3 ChIP at the maternal and paternal *Kcnq1ot1* ICR and imprinted gene promoters in control and *Nup*-depleted XEN cells (*n* = 3; *n* = 3 technical replicates). **b** H3K9me2 and H3K27me3 ChIP at the maternal and paternal *Kcnq1ot1* ICR and imprinted gene promoters in control and *Nup*-depleted XEN cells (*n* = 3 biological samples with three technical replicates per sample). The *y*-axis indicates total and allelic ChIP enrichment levels represented as percent of input. Allelic proportions are represented as a percent of the total ChIP enrichment level. Error bars, s.e.m.; *, significance *p* < 0.05 of paternal/maternal levels compared to the siNT paternal/maternal control (*t* test)

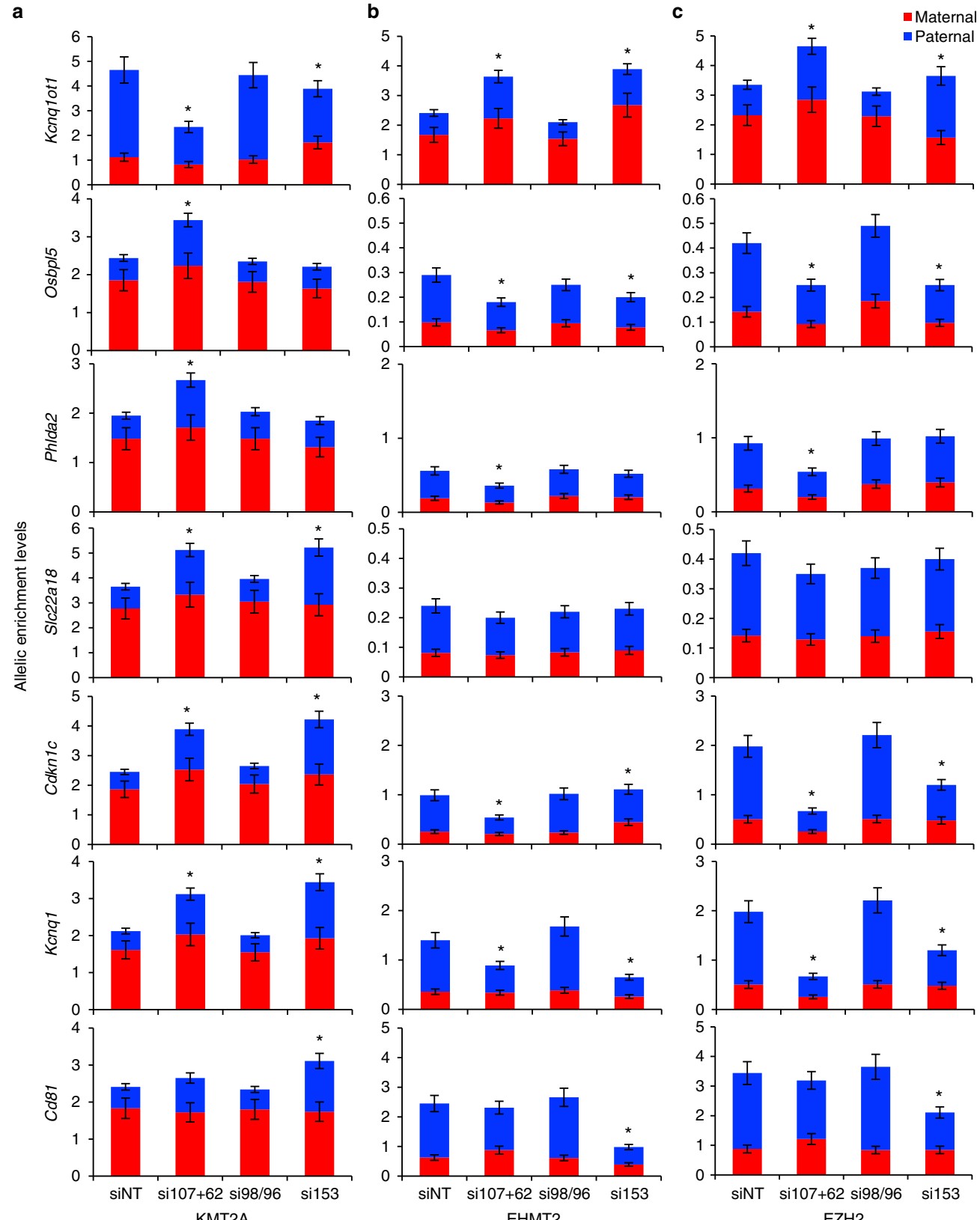

**Fig. 6** *Nup107, Nup62,* and *Nup153* depletion disrupts KMT2A, EHMT2, and EZH2 occupancy at the *Kcnq1ot1* ICR and imprinted gene promoters. **a** KMT2A, **b** EHMT2 and **c** EZH2 ChIP at the maternal and paternal *Kcnq1ot1* ICR and imprinted gene promoters in control and *Nup*-depleted XEN cells (*n* = 3 biological samples with three technical replicates per sample). The *y*-axis indicates total ChIP allelic enrichment levels represented as percent of input. Allelic proportions are represented as a percent of the total ChIP enrichment level. Error bars, s.e.m.; *, significance *p* < 0.05 compared to the siNT control (*t* test)

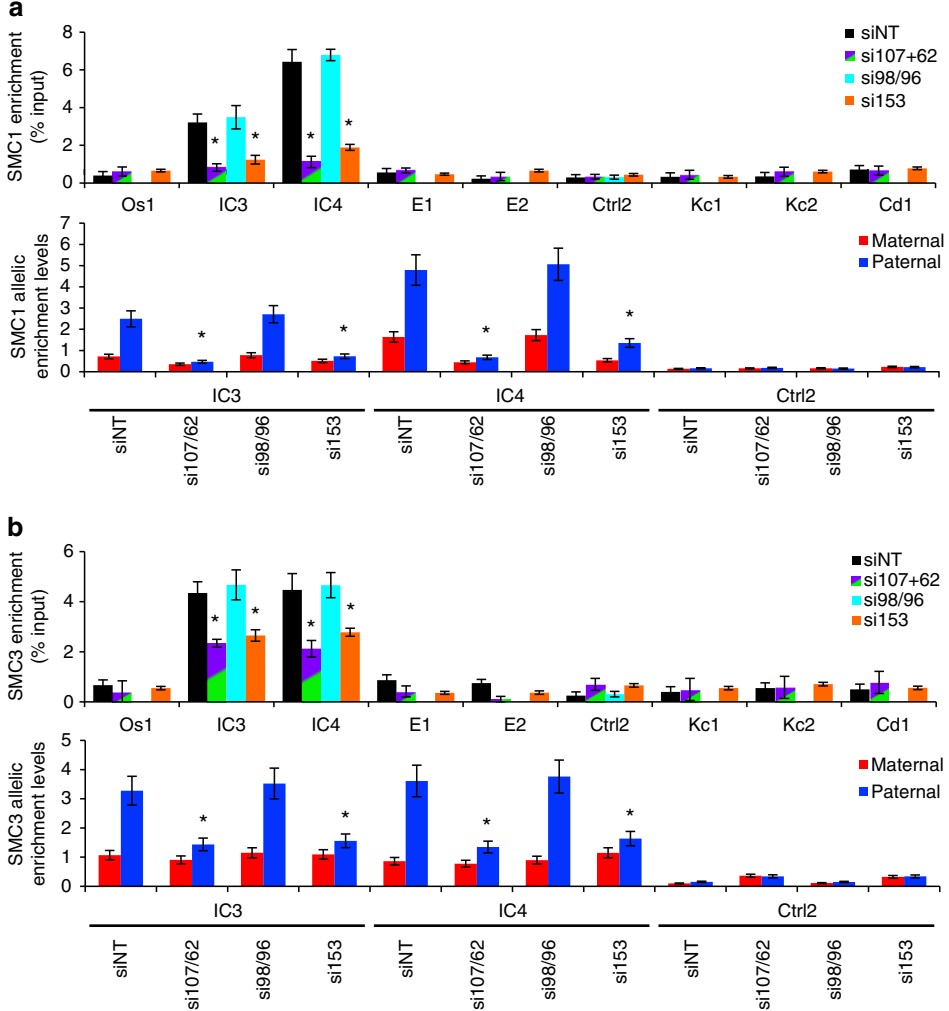

**Fig. 7** SMC1A and SMC3 enrichment at the paternal *Kcnq1ot1* ICR reduced upon nucleoporin depletion. Quantitative ChIP analysis and allelic analysis for **a** SMC1A and **b** SMC3 at positive mAb414 and NUP153 enrichment sites in control and *Nup107/Nup62*-double, *Nup98/96*- (IC3, IC4 only) and *Nup153*-depleted XEN cells (*n* = 3 biological samples with three technical replicates per sample). Allelic proportions are represented as a percent of the total enrichment level. Error bars, s.e.m.; *, significance *p* < 0.05 compared to the siNT control (*t* test)

CTCF, SMC1A, and SMC3, indicating that CTCF and cohesin interactions are mediated through NUP107 (Fig. 8b).

## Discussion

In recent years, studies in mammalian cells, yeast, and *Drosophila* have identified roles for nucleoporins in gene regulation[22,23,27,28,46,47], although the mechanistic action of nucleoporins in this regulation is not fully understood. Here, we identified a nucleoporin-dependent regulatory mechanism at the *Kcnq1ot1* imprinted domain. More specifically, NUP107, NUP62, and NUP153 regulated paternal *Kcnq1ot1* domain positioning at the nuclear periphery; *Kcnq1ot1* ncRNA expression and volume; paternal allelic repression of multiple imprinted genes; maintenance of an active conformation at the paternal *Kcnq1ot1* ICR and a repressed conformation at the paternal *Osbpl5*, *Phlda2*, *Slc22a18*, *Cdkn1c*, *Kcnq1*, and *Cd81* promoters; and cohesin complex interactions at the paternal *Kcnq1ot1* ICR (Supplementary Fig. 28).

Interactions between NUP107, NUP62, and NUP153 have previously been documented in HeLa cells[35,37,48,49]. Investigation of various nucleoporin interactions in U2OS cells found that three of the 11 tested had the capacity to recruit other

nucleoporins, which included NUP153 and NUP107[50]. The NUP153- and NUP107-induced structure also repositioned an integrated chromatin marker from the nuclear interior to the nuclear periphery. By comparison, NUP98 possessed very limited capacity to recruit nucleoporins or target chromatin to the nuclear periphery. In keeping with this, we found that *Nup107*, *Nup62*, and *Nup153* depletion diminished *Kcnq1ot1* domain positioning at the nuclear periphery, suggesting that NUP107, NUP62, and NUP153 act at the *Kcnq1ot1* ICR to tether the *Kcnq1ot1* domain to the nuclear periphery. Coordinate binding of NUP107 with NUP62 and NUP153 likely mediates their regulatory role in *Kcnq1ot1* ICR. Having said this, other sites within the domain maintained their NUP107/62 and NUP153 enrichment upon *Nup153* depletion or *Nup107/Nup62* depletion, respectively. For example, we observed that NUP107/62 remained bound at the *Osbpl5* promoter and enhancer upon *Nup153* depletion. However, even when these nucleoporin interactions were maintained, the *Kcnq1ot1* domain shifted away from the nuclear periphery. The most direct explanation is that NUP107/62 and NUP153 coordinately tethered the *Kcnq1ot1* ICR to the nuclear periphery through the nuclear pore complex, while NUP107/62 and NUP153 binding at other regions of the domain

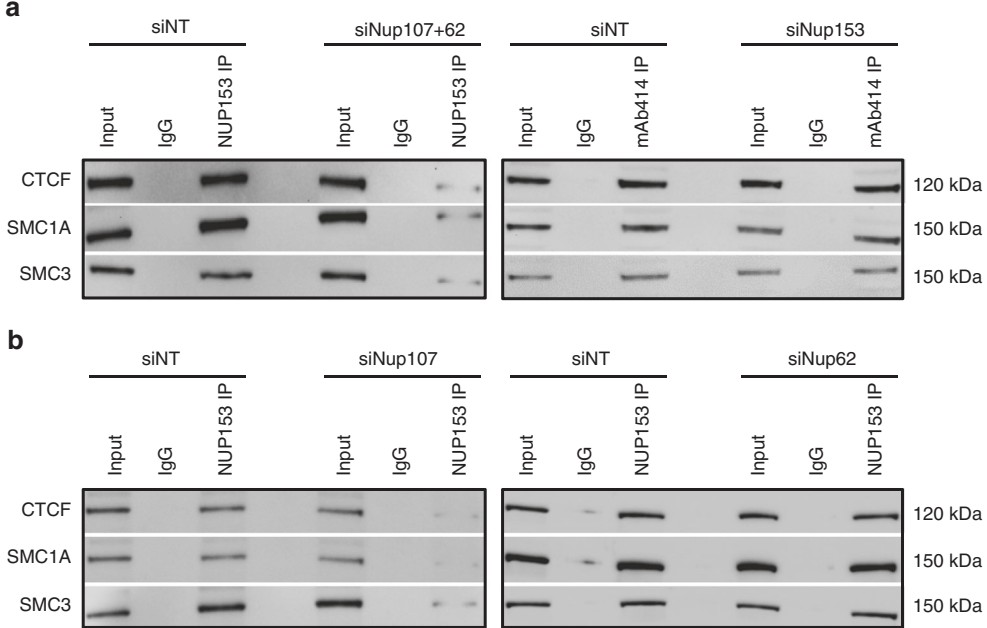

**Fig. 8** NUP107 is required for NUP153 and NUP62 interaction with CTCF, SMC1A, and SMC3. **a** NUP-immunoprecipitation with NUP153 and mAb414 antibodies was performed on control (siNT) and *Nup107/62*- and *Nup153*-depleted XEN nuclear lysates, following which western analysis was conducted using CTCF, SMC1A, and SMC3 antibodies (n = 3 biological replicates). **b** NUP-immunoprecipitation with NUP153 antibodies was performed on *Nup107*- and *Nup62*-depleted XEN lysates, followed by western analysis using CTCF, SMC1A, and SMC3 antibodies (n = 3 biological replicates)

were through an off-pore function. Recent studies have identified multiple nucleoporins (NUP62, NUP153, NUP98 and NUP50) that are present in the nucleoplasm as mobile fractions and regulate genes away from the nuclear periphery by binding their promoters[27,51–53].

Previous studies demonstrated that the *Kcnq1ot1* ncRNA-coated domain is compartmentalized to the nuclear periphery or perinucleolar regions[7,8,29], which suggests a linkage between the *Kcnq1ot1* ncRNA and nuclear positioning. Here, while *Nup107*, *Nup62*, and *Nup153* depletion displaced nuclear peripheral positioning of the paternal *Kcnq1ot1* domain, the *Kcnq1ot1* ncRNA remained associated with the *Kcnq1ot1* domain in *Nup107*-, *Nup62*-, and *Nup153*-depleted XEN cells, albeit with lower *Kcnq1ot1* volumes. This suggests that the *Kcnq1ot1* ncRNA per se does not regulate paternal *Kcnq1ot1* domain positioning at the nuclear periphery. Alternatively, *Kcnq1ot1* ncRNA coating must reach a specific threshold level to enable nuclear periphery positioning. However, the shift from random to nuclear periphery positioning of the maternal *Kcnq1ot1* domain (low *Kcnq1ot1* ncRNA volumes) in *Nup153*-deleted cells does not support the idea of threshold levels.

Previous studies found domain-wide loss of paternal allelic silencing in midgestation embryos harboring paternal deletion of the *Kcnq1ot1* ICR or truncation of the paternal *Kcnq1ot1* ncRNA[5,6,12]. This contrasted with our findings in *Nup107*-, *Nup62*-, and *Nup153*-depleted XEN cells, where paternal alleles were reactivated at a core group of imprinted genes (*Slc22a18*, *Cdkn1c*, and *Kcnq1*), and at specific imprinted genes in a nucleoporin-specific manner (*Osbpl5* in *Nup107 and Nup62*, *Phlda2* in *Nup107*, and *Cd81* in *Nup153*-depleted cells). One possible explanation for lack of domain-wide loss of paternal allelic silencing is that the *Kcnq1ot1* ncRNA acts to coordinately regulate paternal allelic silencing of the core group of genes, while genes more distal from the *Kcnq1ot1* ICR were regulated by a *Kcnq1ot1* ncRNA-independent mechanism. Alternatively, *Kcnq1ot1* ncRNA may operate in a gene-specific capacity across the domain. In day 14.5 placenta, *Kcnq1ot1* ncRNA interactions

were found at genes within (*Cdkn1c* and *Kcnq1*) and outside (*Osbpl5* and *Cd81*) the core group[8]. Interactions between the *Kcnq1ot1* ncRNA and PRC2 or EHMT2 could direct repressive epigenetic marks, H3K27me3 and H3K9me2, respectively, to these paternally silent genes[8]. This is supported by our data, since alterations in KMT2A, EHMT2, and EZH2 binding occurred concomitantly with changes in H3K4me3, H3K9me2, and H3K27me3 levels, respectively, at the *Kcnq1ot1* ICR and paternal imprinted gene promoters upon *Nup107/62* and/or *Nup153* depletion. Recently, NUP153 was identified to play a role in PRC1 silencing of developmental genes in ES cells[27]. This is consistent with reactivation of the normally silent maternal *Kcnq1ot1* allele upon *Nup153* depletion. Thus, nucleoporins may interact with different macromolecular epigenetic complexes to regulate gene expression. For example, NUP107, NUP62, and NUP153 may interact with a complex containing KMT2A to enable *Kcnq1ot1* ncRNA expression, while NUP153 may engage with a repressive complex to regulate maternal *Kcnq1ot1* ncRNA silencing.

A coordinate role for NUP107 and NUP62 has been identified at spindles and kinetochores[54], where they may function in chromatin reorganization upon nuclear membrane reformation[23]. Our findings also point to a role for NUP107, NUP62, and NUP153 in conferring chromatin structure at the *Kcnq1ot1* domain. Interactions between NUP107, NUP62, and NUP153 and the cohesin complex at the nuclear periphery on both sides of the *Kcnq1ot1* promoter could enable an open chromatin configuration at the paternal *Kcnq1ot1* ICR, enabling *Kcnq1ot1* transcription. This is supported by reactivation of the maternal *Kcnq1ot1* ncRNA allele coincident with a shift in maternal *Kcnq1ot1* domain positioning at the nuclear periphery upon *Nup153* depletion. Demarcation of active chromatin by NUP107, NUP62, NUP153 and cohesin may separate active chromatin at the paternal *Kcnq1ot1* ICR from repressive chromatin at the paternal alleles of the core group of imprinted genes, blocking their expression. Upon *Nup107/Nup62* and *Nup153* depletion, cohesin binding was reduced at the paternal *Kcnq1ot1* ICR,

possibly altering chromatin states at the paternal *Kcnq1ot1* ICR and the core group of imprinted genes, leading to alteration in paternal allelic expression. In addition, NUP107/NUP62 binding at the *Osbpl5* promoter, and NUP153 binding at the *Cd81* promoter could demarcate repressive chromatin at *Osbpl5* and *Cd81* from active chromatin at neighboring nonimprinted genes in XEN cells. Upon *Nup107/Nup62* and *Nup153* depletion, chromatin structure change at the paternal *Osbpl5* and *Cd81* promoters could lead them to adopt the active chromatin structure of neighboring nonimprinted genes. Thus, further investigations are required to determine the role of NUP107, NUP62, and NUP153 in regulating higher-order chromatin structure at the *Kcnq1ot1* imprinted domain.

Overall, dynamics of nucleoporin-mediated regulation of the *Kcnq1ot1* domain highlights the complexity of epigenetic processes required to maintain imprinting across a large imprinted domain during early development. Further investigations are required to determine whether this mechanism in XEN cells is conserved in embryonic and trophoblast stem cells. The question also remains whether nucleoporins act as global imprinting regulators, operate at a subset of imprinted domains, or are specific to the *Kcnq1ot1* imprinted domain. As a preliminary query, we analyzed NUP153 DAM ID-seq from ES cells, and found enrichment peaks at multiple ICRs, including *Gtl2*, *Snrpn*, *Igf2r*, *Peg1* and *Peg10*[27], in addition to the at *Kcnq1ot1* ICR[27]. This suggests a more generalized role for NUP153 and possibly other nucleoporins in imprinted domain regulation. Finally, identification of additional candidate epigenetic factors involved in paternal allelic silencing at the *Kcnq1ot1* domain opens a new dimension of investigations into understanding imprinted domain regulation.

## Methods

**XEN cell culture, treatments, and transfection.** CASTX*Cdkn1c*$^{\Delta neoR}$ B6 XEN stem cells were generated by crossing *a Mus musculus casteneus* (CAST) female (Jackson Laboratory) to a C57BL/6J (B6) male carrying a targeted mutation of the *Cdkn1c* gene in which exons 1 and 2 were replaced by the PGK-Neomycin resistance cassette (*Cdkn1c*$^{\Delta neoR}$)[18] (Jackson Laboratory). B6XCAST XEN cells were generated from crosses between C57BL/6J (B6) female and CAST males. Briefly, embryos were collected at embryonic day E3.5 as late morula or blastocyst and plated on mitomycin C (Sigma, M4287)-treated mitotically inactive MEFs in XEN cell medium[17] (RPMI (Sigma, R0883) supplemented with 50 μg/mL penicillin/streptomycin, 1 mM sodium pyruvate, 100 μM β-mercaptoethanol, 2 mM L-glutamine, 15% hyclone ESC grade FBS). On day 2, blastocysts hatched and attached to the plate. On days 3–4, small outgrowths were formed for each embryo. XEN cell medium was replaced every 2 days until XEN cell colonies appeared, which were further passaged on mitomycin C mitotically inactive MEFs. CASTX*Cdkn1c*$^{\Delta neoR}$ B6 XEN stem cells were genotyped for paternal *Cdkn1c*$^{\Delta neoR}$ allele using p57neo primers (Supplementary Table 1). Experiments were performed in compliance with guidelines set by the Canadian Council for Animal Care, and the policies and procedures approved by the University of Western Ontario Council on Animal Care. XEN cells were cultured on a gelatin-coated (EmbryoMax® 0.1% Gelatin Solution, Millipore, ES-006-B), feeder-free environment prior to experiments to avoid feeder contamination[1].

Three wild-type B6XCAST XEN cell lines, which were produced from different embryos, with 2–5 technical replicates per line, were used for each experiment, except for the DNA methylation analyses, where two biological replicates were analyzed. Where specified, cells were synchronized in G1 phase by treatment with 2 mM hydroxyurea (Sigma, H8627) for 12 h followed by siRNA transfection. Cells were transfected with 10 nmol of siRNAs with 3.6 μL siPepMute™ siRNA Transfection Reagent (SignaGen, SL100566) in 100 μL of 1× transfection buffer. Fresh XEN medium was added 2 h before transfection. siRNA sequences are listed in Supplementary Table 2.

**Recombinant virus production and lentiviral transduction.** A library of shRNAs targeting 250 different epigenetic factors used in this experiment was previously described[17]. Transduction of shRNAs into XEN cells was modified from ref. [17]. Briefly, shRNA vectors and lenticlass plasmids encoding Vesicular stomatitis virus glycoprotein pseudotype (plasmid pMDG) and Psi (pPsi) packaging elements were transfected into 10 cm dishes of HEK293 cells (ATCC®, CRL-3216™) using Lipofectamine 2000 (Invitrogen, 11668027). The ratio of plasmids transfected was 2.3 μg shRNA vectors, 1.0 μg pMDG, and 1.7 μg of pPsi. Medium was changed 6 h later. Viral-containing media was harvested and filtered (0.45 μm pore) 72 h post-

transfection and added to freshly trypsinized XEN cells along with 1× polybrene (Sigma, TR-1003-G) into 1× 6-well plates (Corning, C3506). XEN cell media was changed 48 h later. Low levels of infection were necessary to ensure single viral integration per cell and was verified using flow cytometry. For flow cytometry analysis, cells were washed twice with 1× PBS (Sigma, D8537) and trypsinized (Sigma, T4174) into single cell suspensions. Cells were pelleted and resuspended in 10% FBS in 1× PBS solution followed by analysis on Beckman Coulter Epics XL-MCL Flow Cytometer. XEN cells were selected for viral integration with 1 μg/mL puromycin (Sigma, P8833). shRNA containing XEN cells were moved to 8 × 10 cm dishes (Corning, 430167) and selected for loss of silencing of the *Cdkn1c*$^{\Delta neo}$ allele with 50–200 μg mL$^{-1}$ neomycin (Sigma, G8168). While still under neomycin selection, XEN cells were replated at a lower density onto 15 cm dishes until individual colonies were formed. Individual colonies were picked and transferred over to 24-well dishes (Corning, C3527), and then underwent a second round of neomycin selection.

**DNA and RNA isolation, cDNA preparation and PCR amplification.** DNA of surviving colonies from the lentiviral screen was isolated using Qiagen DNeasy Blood & Tissue Kit (Qiagen, 69504) and PCR amplification was performed for the hairpin region inside the shRNA construct with primers targeting the lentiviral sequence[17]. RNA was isolated using PureLink® RNA Mini Kit (Life Technologies, 12183018A), QuickRNA™ MicroPrep (Zymo Research, R1050) or RNeasy Plus Mini Kit (Qiagen, 74134) according to the manufacturers' instructions. Before cDNA preparation, total RNA was subjected to DNase I (Life Technologies, 18068015) treatment as described[1]. cDNA was prepared using ProtoScript® II Reverse Transcriptase (NEB, M0368X) as per instructions with oligodT$_{23}$ (Sigma, O4387) and Random Primers (Life Technologies, 48190011). cDNA was treated with RNaseA (Roche, 10109142001) after preparation to remove residual RNA. PCR was performed on CFX1000 and MJ Research Thermocyclers (Bio-Rad).

**Quantitative PCR analysis.** Quantitative (q) PCR was performed as described[1]. Briefly, qPCR was performed using iQ SYBR Green Supermix (Bio-Rad, 1708880) or SensiFAST™ SYBR® No-ROX Kit (Bioline, BIO-98005) on an MJ Thermocycler Chromo4 Real-time PCR system (Bio-Rad, CFB-3240) or a CFX-96™ Real-time system (Bio-Rad, 185-5096). For gene expression analysis, data were analyzed using the 2ΔΔCT method[55]. For ChIP, qPCR data analysis was performed as described[56]. Supplementary Table 1 lists primers, annealing temperatures, and amplicon sizes.

**Droplet digital PCR allelic expression analysis.** Droplet digital PCR (ddPCR) probes and primers were designed following Bio-Rad's guidelines for ddPCR assays. Single nucleotide polymorphisms (SNPs) between C57BL/6 and CAST genomes at target genes were assessed with OligoAnalyzer 3.1 [57]. SNPs that generated the greatest difference in melting temperature ($T_m$) between the alleles, in either forward and reverse compliment strands, were selected as the probe target. Primers and probes were designed using Primer3plus[58], RealTimeDesign™ software (BIOSEARCH Technologies)[59], OligoAnalyzer 3.1 [57], or advice from IDT. For primer design, a target $T_m$ was set for ~60 °C with each primer being within ~2 °C of each other. When possible, amplicon size was kept between 80 and 200 bp, but larger amplicons were possible with optimized PCR amplification programs. Amplicons with repeats of Gs and Cs longer than three bases, especially at the 3′-end, were avoided. For probe design, the target length was set to 11–22 bp, and where possible the SNP was centered within the probe. Caution was taken to ensure that a G nucleotide was not at the 5′ of start of probe. A probe $T_m$ of 3–10 °C higher than the primer pairs was required. Shorter probes incorporated locked-nucleic acids (LNA) to achieve the probe target $T_m$. EXIQON LNA Oligo Tm Prediction[60] was used to predict $T_m$ changes from adding LNAs. Increased probe $T_m$ were also achieved with longer probes, in this case a ZEN-quencher (Integrated DNA Technologies; IDT) was added to the center of the probe to dual-quench the fluorophores. Following probe design, a $T_m$ difference of ≥3 °C between the probes was confirmed using OligoAnalyzer 3.1 [57]. Probes for *Cd81*, *Cdkn1c*, *Th*, and *Tssc4* were designed by IDT. All probes were synthesized by IDT with FAM or HEX fluorophores quenched by Iowa Black FQ. Primer and probe sequences are listed in Supplementary Table 1.

For ddPCR sample preparation, RNA was extracted using the RNeasy Plus Mini Kit (Qiagen, 74134) according to the manufacturers' instructions. RNA concentrations were determined using Qubit® 3.0 Fluorometer (Life Technologies, Q33216), Qubit RNA BR Assay Kit (ThermoFisher Scientific, Q10210) and Qubit RNA HS Assay Kit (ThermoFisher Scientific, Q32852). Before cDNA preparation, total RNA was subjected to DNase I (Life Technologies, 18068015) treatment. cDNA was prepared using ProtoScript II Reverse Transcriptase (NEB, M0368X), oligo(dT)$_{23}$ (Sigma, O4387) and Random Primers (Life Technologies, 48190011). cDNA was treated with RNaseA (Roche, 10109142001) after preparation to remove residual RNA. For each gene, ddPCR master mixes were made for a reaction final volume of 22 μL, containing; 2–4 μL cDNA, 1 μL of primers each (250 nM), 1 μL of probes each (FAM, HEX 250 nM), 11 μL ddPCR Supermix for Probes (no dUTP) (Bio-Rad,1863024), and 3–5 μL DNASe-RNase free UltraPure dH$_2$0 (Invitrogen, 10977-015). Each reaction was loaded into a 96-well plate (Bio-Rad, 12001925), sealed using a foil heat seal (Bio-Rad, 1814040) and placed into a PX1 PCR Plate Sealer at 180 °C for 5 s. Sealed plates were loaded into the AutoDG (Bio-Rad,

1864100) for droplet generation. Droplets were loaded into a new 96-well plate (Bio-Rad, 12001925) and sealed using a foil heat seal (Bio-Rad, 1814040). Droplet digital PCR (ddPCR) was performed on a deep well CFX1000 (Bio-Rad,1851197). Optimal annealing temperature for each primer/probe set is listed in Supplementary Table 1. For most genes, a standard ddPCR program of 10 min at 95 °C, 40 cycles of 94 °C for 30 s and annealing temperature for 60 s, and a final cycle of 10 min at 98 °C. For *Cdkn1c*, that has a longer amplicon, an additional step was added to the program to enhance fluorophore release (10 min at 95 °C, 40 cycles of 94 °C for 30 s and 60 °C for 60 s, 10 cycles of 94 °C for 30 s and 50 °C for 60 s, and a final cycle of 10 min at 98 °C). Following PCR, the droplets were read on the QX200 (Bio-Rad, 1864100), set for detecting absolute levels of FAM/HEX probe fluorophores. Each positive droplet in a ddPCR assay represents a template molecule in the sample, allowing for absolute quantification of allelic expression. Using Quantasoft Analysis Pro 1.0.596 (Bio-Rad), droplets from FAM and HEX identified alleles were selected using the 1D and 2D Amplitude tool. Any FAM/HEX double-positive droplets represent doublets containing both alleles, and these were eliminated from analysis. Copies per µL values that were determined from Quantasoft Analysis Pro were converted to copies per µg and plotted using BoxPlotR[61].

**RNA/DNA FISH, immunocytochemistry and confocal microscopy.** *Kcnq1ot1* RNA/DNA FISH probe was generated using a 32-kb fosmid Wl1-2505B3 in the *Kcnq1* intronic region (BACPAC Resources Center at the Children's Hospital Oakland Research Institute, Oakland, CA) using the BioPrime DNA labeling System (ThermoFisher Scientific, 18094011) with fluorescein-12-dUTP (Roche, 11373242910) and Biotin-16-dUTP (Roche, 11093070910) for RNA FISH, and Cy5-UTP (GE Healthcare, 45001239) for DNA FISH as described[1]. For RNA FISH, cells seeded on coverslips (VWR, 48380046) in 12-well plates (Corning, C3512) were fixed, dehydrated in a series of 25, 50, 75, and 100% ethanol followed by hybridization with RNA-FISH probes in 100% molecular grade formamide (VWR, 97062-006) as described[1]. Briefly, following overnight incubation at 37 °C in a formamide sealed hybridization chamber, coverslips were carefully washed in 4× SSC and 2× SSC (ThermoFisher Scientific, AM9763) at 37 °C for 5 min three times each. All washes were done in 12-well tissue culture plates. Coverslips were incubated with primary antibody in blocking buffer (20× SSC, 10% Tween-20 (Sigma, P9416), 10% Skim Milk) at 37 °C for 1 h in the dark, then washed three times with 4× SSC (37 °C) with agitation. Next, coverslips were incubated with the secondary antibody in blocking buffer at 37 °C in the dark for 1 h, then washed three times with 4× SSC and two times with 2× SSC buffers at 37 °C with continuous agitation, 5 min each. The final wash was done in 1× SSC for 5 min. SSC buffer was made fresh from 20× stock and pre-heated to 42 °C during the washes. Coverslips were mounted on glass slides with Vectashield DAPI (antifade mounting medium, Vector labs, H-1000) and stored in the dark for a few hours or overnight at 4 °C. For DNA/RNA FISH, DNA FISH was first performed as described[33] followed by RNA FISH. Briefly, fixed cells were rinsed with 1× PBS and dehydrated in a series of ethanol washes. DNA was denatured by incubating the coverslips with 70% formamide in 2× SSC at 85 °C for 20 min. Coverslips were quickly washed with 2× SSC and hybridized with DNA FISH probes overnight at 37 °C in a sealed formamide chamber (sealed with parafilm). Next day, excess probe was washed with pre-warmed (42 °C) 4× SSC and 2× SSC buffer three times each. Coverslips were then incubated with RNA FISH probe. RNA FISH was performed as described above.

For immunocytochemistry assays, XEN cells were seeded on gelatin-treated coverslips in 12-well plates. Cells were permeabilized with sequential transfers into ice-cold cytoskeletal extraction buffer (CSK) for 30 s, ice-cold CSK containing 0.25% Triton X-100 for 45 s and ice-cold CSK for 30 s. Following CSK treatment, cells were fixed in freshly prepared 4% paraformaldehyde (Sigma, P6148) at room temperature for 10 min, washed twice in 1× PBS and stored in 70% ethanol for up to a month. Cells were washed three times in 1× PBS for 5 min each followed by incubation in primary antibody in 5% skim milk for 1 h at room temperature, washed three times with 1× PBS followed by incubation with secondary antibody. Cells were then washed three times with 1× PBS, stained with DAPI and mounted on glass slides for confocal microscopy. Supplementary Table 3 lists antibody dilutions.

Coverslips were imaged using z-stacks on a FluoView FV1000 coupled to an IX81 motorized inverted system (Olympus). For DNA and RNA FISH, signal volume was measured in Volocity (PerkinElmer) with z-stacks intensity-based thresholding (µm³), while distances of DNA FISH signal centroid from the nuclear rim were calculated to determine domain positioning (µm) (ImageJ).

**Western blot analysis and immunoprecipitation assay.** Cytoplasm and nuclear protein extracts were isolated using cytoplasm extraction buffer and nuclear extraction buffer using the NE-PER Nuclear and Cytoplasmic Extraction Kit (ThermoFisher Scientific, 78833) followed by western blot analysis as described[17,62]. Briefly, protein samples were quantified using the Bradford assay, Nanodrop or Qubit protein concentration determination assay. Typically, 20–30 µg of protein was separated on an 8–12% SDS-PAGE (Bio-Rad, Bulletin 6201) gel or any kD™ Mini-PROTEAN® TGX™ Precast Protein Gels (Bio-Rad, 4569033), transferred to PVDF membrane (Bio-Rad, 1620177) or Trans-Blot® Turbo™ Mini PVDF Transfer Packs (Bio-Rad, 1704156), blocked for 1 h in 5% skim milk in 1× TBST (1× TBS (Bio-Rad, 1706435) with 0.1% Tween-20, (Sigma, P9416)), incubated with primary

antibody in 5% skim milk as described in Supplementary Table 3 followed by three washes of 1× TBST for 7 min each. Blots were next incubated with secondary antibody in 5% skim milk followed by detection using chemiluminescence reaction with Clarity Western ECL Substrate (Bio-Rad, 1705060). For list of antibodies and dilutions see Supplementary Table 3. Uncropped scans of western blots presented in the main figures are provided in Supplementary Figs. 30–38.

Protein co-immunoprecipitation was performed using Dynabeads Protein G (Invitrogen, 10004D) as per the manufacturer's instructions. Briefly, 50 µg of nuclear lysates obtained from XEN cells using the NE-PER Nuclear and Cytoplasmic Extraction Kit (ThermoFisher Scientific, 78833) were incubated with the respective antibody, 2 µL RNase A (Roche, 10109142001), 2 µL DNase I (Life Technologies, 18068015), and 4 µL protease inhibitor (Sigma, P8340) as described in Supplementary Table 3 overnight at 4 °C. Samples were then incubated with Dynabeads protein G (Invitrogen) for 1 h at 4 °C. Immunoprecipitates were washed three times with 200 µL of 0.1% Tween-20 in 1× PBS, eluted in 1× Laemmli buffer (Bio-Rad, 1610747) and resolved on 10–12% SDS-PAGE followed western blot analysis as described above.

**Chromatin immunoprecipitation (ChIP) assay.** ChIP was performed as described with modifications for siRNA depletions[56]. Equal numbers of cells were collected from control and siRNA treatment groups (~0.5–1.2 million cells). Approximately 40,000 cells were used per ChIP reaction. Briefly, cells were cross-linked in freshly prepared 1% formaldehyde final concentration for 6 min, quenched in 1.25 M glycine at room temperature for 5 min, followed by wash in 200 µL 1× PBS. Pellets were resuspended in 200 µL of cytoplasm extraction buffer (10 mM HEPES (pH 7.6), 60 mM KCl, 1 mM EDTA, 1 mM DTT, 0.075% NP40, 1 mM PMSF) and incubated on ice for 5 min followed by spin at 3000 rpm at 4 °C for 4 min. Nuclear pellets were lysed by resuspension in 100 µL 1% SDS buffer for 10 min on ice, followed by addition of 100 µL 1× TE, 3µL of proteinase inhibitor (Sigma, P8340) and then sonicated (Diagenode, B01060001 or Covaris M220) to shear chromatin into 100–400 bp fragments, followed by immunoprecipitation. Frozen cross-linked chromatin lysates were stored at −80 °C. Protein G Dynabeads (8 µL, Life Technologies) were washed three times with ChIP dilution buffer (0.1% SDS, 1% Triton X-100, 2 mM EDTA, 20 mM Tris pH 8.1, 500 mM NaCl). Magnetic beads were bound with 1.5–4 µg of antibody at 4 °C for 1 h followed by incubation with precleared chromatin lysate (equivalent to 40,000 cells) for 3 h to overnight (as specified in antibody table). Chromatin lysate (10 µL) was saved as Input. Beads were washed at 4 °C once with 200 µL of Low Salt Buffer (0.1% SDS, 1% Triton X-100, 2 mM EDTA, 20 mM Tris pH 8.1, 150 mM NaCl) for 5 min, followed by two washes with 200 µL of High Salt Buffer (0.1% SDS, 1% Triton X-100, 2 mM EDTA, 20 mM Tris pH 8.1, 500 mM NaCl) for 5 min each. Beads were next washed once with 200 µL 1× TE Wash Buffer (10 mM Tris pH 8.1, 1 mM EDTA) at 4 °C for 5 min. DNA was eluted off the beads by incubation at 65 °C for 1 h with constant shaking in 200 µL of elution buffer (1% SDS,100 mM NaHCO₃). To purify eluted DNA, RNA was degraded by addition of 1 µL of 1 µg/µL RNase A (Roche) with an incubation at 37 °C for 1 h. Protein was degraded by addition of 4 µL of 10 mg/mL proteinase K (Sigma) with an incubation at 55 °C for 2 h. Following ChIP, DNA extraction was performed using Chelex beads (Bio-Rad, 1421253), ChIP DNA Clean & Concentrator™ (Zymo Research, D5201) or QIAquick PCR Purification Kit (Qiagen, 28104). ChIP-qPCR assays were performed using iQ SYBR Green Supermix (Bio-Rad, 1708880) and SensiFAST™ SYBR® No-ROX Kit (Bioline, BIO-98005) on an MJ Thermocycler Chromo4 Real-time PCR system or a CFX-96 Real-time system (Bio-Rad) under the following conditions: 95 °C for 5 min followed by 40 cycles of 95 °C for 10 s, annealing (for temperature see Supplementary Table 1) for 20 s, 72 °C for 30 s followed by plate reading. Lastly, a final melting curve was generated from 55 to 95 °C in increments of 1 °C. Enrichment levels (percent input) were calculated $100 \times (2^{[\Delta Ct_{Input} - \Delta Ct_{Input}]} - [\Delta Ct_{Input} - \Delta Ct_{Ab}])/25$. Supplementary Table 3 lists antibody concentration. Data are represented as total enrichment (percent input) or allelic enrichment (calculated as total enrichment × allelic enrichment ratio).

**Statistical analysis.** For statistical analysis, two-tailed Student's *t* test was performed on mean values. Treatment samples were compared to controls. A *p* value less than 0.05 was considered to be significant.

**Data availability.** All data generated or analyzed during this study are included in this published article (and its supplementary information files). Data supporting the findings of this manuscript are available from the corresponding author upon reasonable request.

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

## Acknowledgements

We thank Fred Dick for helpful comments, Rashid Mehmood for the E47-RFP[NLS] construct, and Erik Wendlandt, IDT, for assistance with droplet digital PCR probe design. This work was supported by grants from the Canadian Institutes of Health Research (MOP 111210), Western University, Schulich School of Medicine and Lawson Health Research Institute, Magee-Womens Research Institute and the University of Pittsburgh to M.R.W.M. S.S.S. was supported by Western University Obstetrics and Gynaecology Student Fellowship, Children's Health Research Institute Curtis Cadman Studentship, Lawson Health Research Institute Graduate Fellowship, and Magee-Women's Research Institute Paul M. Rike Postdoctoral Fellowship. L.S.L. was supported by an NSERC Canada Graduate Scholarship. C.R.W. was supported by Ontario Graduate Scholarship. W.A.M. was supported by Children's Health Research Institute Postdoctoral Fellowship. M.R.W.M. is a Magee Auxiliary Research Scholar.

## Author contributions

S.S.S. performed experiments and wrote manuscript. L.S.L. performed shRNA screen. L.Z. performed Western analysis, immunoprecipitation, and provided technical assistance for ChIP analysis. C.R.W. performed bisulfite mutagenesis assay. W.A.M. designed droplet digital assays and provided bioinformatics analysis. M.C.G. designed and supervised the lentiviral screen. M.R.W.M. designed and supervised the project, helped with data interpretation and manuscript preparation. All authors have read and approved the final manuscript.

## Additional information

**Competing interests:** The authors declare no competing interests.

