## [Peer Review File · Nature Communications]

Reviewers' comments:

Reviewer #1 (Remarks to the Author):

Genomic imprinting is one of the most well-studied epigenetic systems, which is mammal-specific monoallelic gene expression. Recently, thanks to great advance in sequencing technologies, many important studies regarding the regulatory mechanism and life cycle of imprinting have been reported. However, how monoallelic expression is maintained through development, especially during global DNA demethylation at preimplantation stage, remained unclear. Although some factors including Zfp57 and Kap1 have been reported to play major role in imprinting maintenance, it has been suggested that additional unknown factors are involved in this process.

In this manuscript, Sachani et al. addressed this question through RNAi screening using extraembryonic endoderm stem (XEN) cells. Based on this screening, authors identified several candidates, including Ezh1, Smarca5, and some of nucleoporin (NUP) family proteins. Authors more focused on the mechanism of NUP-mediated imprinting regulation, and found that NUP play a role in the regulation of Kcnq1ot1 domain through (1) Kcnq1ot1 expression, (2) sub-nuclear localization of Kcnq1ot1 locus, (3) mono-allelic expression of core imprinted genes in Kcnq1ot1 domain, (4) allele-specific histone modifications, and (5) interaction of cohesion complex at the paternal allele. Although NUP has been reported to regulate gene expression through nuclear architecture, as far as this reviewer knows, contribution of NUP in imprinting maintenance has not been reported yet. In this manuscript, authors carefully addressed the molecular mechanism through knockdown experiments in the B6:CAST hybrid XEN cells. Since imprinting maintenance mechanism is one of the most important question in this field, this reviewer believe this study is worth considered to be published in Nature Communications with some additional experiments.

Major comments

Although authors analyzed the molecular mechanism of NUP-mediated monoallelic expression of Kcnq1ot1 region in detail using XEN cells, critical question to be solved is whether this mechanism is physiologically important through development. To address this question, knockdown (or knockout by CRISPR) experiment in preimplantation embryo should be performed. RT-qPCR and/or allelic expression analysis of some of core imprinted genes in the blastocysts from siRNA-injected zygote or parthenogenetic embryos would be informative and improve the value of this manuscript significantly.

Minor comments

P5 line 93: "while SMARCA5 and SMARCA5 regulation is" should be "while SMARCA5 and SMARCA5 and SMARCA5 regulation is". Related to this title, the conclusion "Smarca5 and Smarcad1 function in a Kcnq1ot1 ncRNA/transcription-dependent manner at the paternal Kcnq1ot1 domain" (P7 line 129-130) is not proved by evidence. What authors found is just 30 to 50% down-regulation of Kcnq1ot1 in Smarca5 and Smarcad1 knockdown cells (Fig. 1b), and this could be coincidence. Without rescue experiments, this conclusion is over-statement.

P7 line 4: "The most unexpected candidate is our screen was" seems to be typo. This reviewer thinks it should be "The most unexpected candidate in our screen was".

P13 line 278" "Supplementary figure 14a" should be "Supplementary figure 14b"

P13 line 283" "Supplementary figure 14" should be "Supplementary figure 14d"

Some of sample numbers mentioned in figure legends are strange. For example, in Supplementary figure 19a and b showing IP-Western blotting results, what "n=2-3; n=3 technical replicates" means? Did authors combine 2 or 3 samples and analyzed together? This sample number information was not shown in Supplementary figure 6. Please clarify them.

Figure 1 legend: Comma and semicolon are confused, "MAT/B6, maternal (red); PAT/CAST;

paternal (blue)" should be "MAT/B6, maternal (red); PAT/CAST, paternal (blue)"

Figure 2(f) legend: Explanation for what NP, SP, and NI stand for should be described. Some of commas and semicolons were confused (e.g. "Nup98 siRNA, si153" should be "Nup98 siRNA; si153").

Figure 6 legend: It is unclear what Y axis indicates. Is this % input? How P values are calculated, comparison of total ChIP OR proportion of maternal and paternal ChIP? Also, it is unclear what "Details as described in Figure 2" means, as neither conventional ChIP nor allele-specific ChIP are shown in figure 2.

Supplementary Figure 2 (B): No explanation about the experimental methods. Although it is assumable this is the result of RT-PCR, authors should specify it clearly.

Reviewer #2 (Remarks to the Author):

The manuscript reports an analysis of the effects of shRNAs that are designed to target selected chromatin binding proteins on transcription of a set of genes near the *Kcnq1* locus. The authors report that shRNAs against a set of proteins alter transcription of different genes in this region. It is not clear if the shRNAs alter allele-specific imprinting or if they affect the expression of both alleles. The following issues should be addressed:

1. The screen that is described is not shown to respond to imprinting. No evidence is provided that the selection target (Neo) is regulated in a parental allele-specific manner in XEN cells. Whereas it is possible that regulators of the selected locus contribute to imprinted gene regulation, it must be clearly stated that the results of this assay do not reflect on the roles of the tested shRNA targets in imprinted gene regulation.
2. The specificity of shRNA depletion of the targeted proteins is not clearly established. Whereas the levels of the proteins are shown to be reduced by the corresponding shRNAs, the effects of the shRNAs on other proteins known to influence imprinting are not tested. This deficiency is aggravated by the fact that the effects of a single shRNA directed against most proteins are characterized (though several shRNAs were used against some proteins in the screen). Additionally, no attempt is made to restore protein expression or to use independent approaches (e.g, genetic mutations) to corroborate the results. This deficiency is particularly serious in the case of the depletion of individual subunits of protein complexes whose disruption could influence the levels of the other subunits.
3. In several experiments, the levels of transcripts are shown as % maternal and % paternal (e.g, Fig. 1, Fig. 2b, Fig. 5). These data are interpreted to show reactivation of the paternal allele, and a role for the protein targeted by shRNA in allele-specific silencing. However, the data do not demonstrate a selective effect on transcription of one allele. The absolute levels of transcripts from each allele should be shown. This would obviate the need to show transcript levels and allele-specific expression separately (e.g. Fig. 1b and 1c).
4. The conclusion that *Ezh1* function versus *Smarca1/5* function does not or does depend on *Kcnq1* transcription is not justified. The fact that *Kcnq1* bulk transcript level does not or does change has no bearing on whether it is not or is required for function.
5. The interpretation that different Nup gene products regulate only the paternal versus maternal *Kcnq1* allele is not justified (Fig. 2a, 2b). It is equally plausible that they function at both alleles, but that other proteins confer paternal-specific expression.
6. The differences in *Kcnq1* RNA "volume" appear to correlate with transcript levels (Fig. 2). Is there any significance to these differences?
7. The bar graphs that plot paternal *Kcnq1* locus locations (Fig. 2f) do not specify the % of cells where the locus was not detected. Differences in detection of the locus in different cell populations can produce the appearance of a change in localization. The probes that were used to determine locus and transcript volumes are not clearly described.

8. The preferential detection of the paternal allele in Nup immunoprecipitates is potentially significant. It is not clear how the data were normalized to account for potential differences in recovery of the paternal and maternal alleles in the ChIP assay. The same concern pertains to the data in Fig. 6 since the recovery of the maternal versus paternal alleles could change in response to siRNA expression and/or transcription of the locus. Input chromatin is not a valid control since recovery in ChIP assays is affected by many factors. The efficiencies of allelic ChIP must be compared with proteins that bind specifically to both alleles with equal efficiency. The cause versus consequence relationship of the histone modifications and transcription is also unclear. If H3K4me3 and H3K9me2 levels were important in determination of the levels of imprinted gene transcription, one would expect that depletion of the corresponding methyltransferases or demethylases would influence the transcript levels.

9. The in vitro binding data lack important controls to determine if Nups bind DNA specifically. No control DNA fragment is used in the EMSA experiment. Since the specific binding sites are not identified and the characteristics of the probes that are used as negative controls in the DNA affinity precipitation experiment, the data are not convincing.

10. The reported interactions between Nups with RNAPII, CTCF, SMC1 and SMC3 are not well established. The IgG control precipitation may not reflect the full range of non-specific and indirect targets that are precipitated by the α -Nup and anti-RNAPII antibodies that are used. A better control is the precipitation of complexes from cells where the Nup that is recognized by the antibody is depleted.

11. The effects of Nup depletion on SMC1 and SMC3 ChIP efficiencies in the IC3/IC4 regions are the most significant findings relating to their potential roles in ICR regulation. To determine the specificity of these effects, critical controls must be performed to establish if SMC1 or SMC3 binding at other loci where they bind specifically are affected by Nup depletion. Determination of the levels of SMC1 and SMC3 proteins does not establish if their overall levels of chromatin binding are altered.

In summary, the manuscript presents a large amount of data related to the regulation of genes near the *Kcnq1ot1* locus. Significant controls are missing that raise questions about the validity of the interpretations. There is also a larger question whether the mechanisms that are investigated pertain to imprinting or if they regulate the genes that are investigated in an allele-independent manner.

Reviewer #3 (Remarks to the Author):

General

In the manuscript Sachani et al. the authors use loss of *Cdkn1c* silencing as a proxy for loss/impaired imprinting at the *Kcnq1ot1* imprinted domain. They identified a number of epigenetic modifiers that reactivated the paternally silenced Neo cassette, which replaces Exons 1 and 2 of the *Cdkn1c* imprinted gene. Some novel candidates included Nup107 prompting the authors to investigate other nucleoporin proteins such as Nup62, Nup 98 and Nup 153.

1. The authors have used XEN cells to identify factors involved in controlling *Kcnq1ot1* imprinted domain but it is not clear if the role of the genes identified is the same in other cell types, can the authors investigate the relevance of their findings in other cell types?

2. Similarly, to further validate a potential role of the nucleoporins in establishing imprinting they should look at other imprinted domains, this would establish either a general role or specific role in imprinting.

3. The first results section describing validation of EZH1, SMARCA5 and SMARCA1 feels incomplete, the authors investigate a potential role in DNA methylation at the ICR site but do not pursue other possibilities. Perhaps the authors could also investigate their role on histone modifications (particularly applicable to EZH1), it is not clear why this was done for the nucleoporins and not for this set of candidates. Related to this the authors discuss SMARCA5, along with CTCF and the cohesin complex, positively regulating transcription of H19. Given this, it

is not clear why the authors would not test the hypothesis that SMARAC5 regulates Kcnq1ot1 transcription in a similar manner.

4. The authors observe a decrease in Kcnq1ot1 expression upon knockdown of Nup107, Nup62 and Nup153. Kcnq1ot1 has been shown to recruit EZH2 and this is believed to be the mechanism by which it establishes imprinting. This could be the potential mechanism through which the nucleoporins regulate allelic silencing at certain imprinted genes. The authors should investigate whether EZH2 (PRC2) recruitment is reduced – which would in turn provide a mechanism for the reduced H3K27 levels observed. The role of Kcnq1ot1 in recruiting should also be included in the discussion.

5. A number of reports have established the role of b-catenin and Wnt signalling in the function of nucleoporins and Kcnq1ot1 transcription. The author should explore this link in their cell types and following point 1, in other cell types too.

6. Kcnq1ot1 has also been shown to help establish silencing at Kcnq1 through mediating intrachromosomal interactions. As Kcnq1 is one of the core genes impacted upon, is this interaction lost?

7. The material and methods could be improved. There is no description of the method used for methylation analysis. For gene expression the $2\Delta\Delta\text{ct}$ method is used not the $\Delta\Delta\text{ct}$ method. It is also implied the authors used a one-tailed T-test for all analysis. For example a one-sided test assumes the authors have made a hypothesis of the direction of gene expression – NUP62, NUP98 and NUP153 were not included in the original screen and so the 'expected' direction of expression changes for genes cannot be known. Therefore a two-tailed test would be more appropriate. I think certain details are missing from the ChIP method – How long was crosslinking carried out for? How much chromatin was used?

Minor comments

The writing style – there appears to be a lack of narrative where statements are made but not elaborated upon. For example results Lines 98/99.

There are a few examples of where the authors have incorrectly used cohesion complex instead of cohesin complex.

Specific

Abstract line 28 depletion repeated

Abstract line 35/36 – the role of nucleoporins is only established for the Kcnq1ot1 imprinted domains in this manuscript – this sentence implies a general mechanism for all imprinted domains.

Introduction line 43 should be domains not domain

Introduction lines 63-65 some protein symbols are missing – in the abstract protein symbols are not included.

Results line 93 SMARCA5 is repeated, should be replaced by SMARCA1.

Results line 134 why is the most unexpected candidate?

Results line 138/139 I assume the siRNA with the best (or possibly most consistent) knockdown was chosen?

Results line 338-340 – Kcnq1ot1 has been shown to regulate H3K27me3 through EZH2 recruitment. Therefore the alterations of the histone modifications at imprinted gene promoters could be as an indirect result of this.

Discussion there is no mention of the implication of altered cohesin complex interaction.

Discussion there is no mention of the role of Kcnq1ot1 ncRNA in regulating PRC2 recruitment to the imprinted domain. Or the role of Kcnq1ot1 ncRNA in chromatin structure at this locus. There is also evidence for recruitment of G9a – given that H3K9me is also altered this warrants discussion.

Figures

Supplementary figure 11 abstract negative not negatives

Response to Reviewers

We thank the reviewers for their time to review our manuscript and for their insightful comments. Below we have addressed their concerns and feel that these revisions have improved the manuscript significantly. Below are point-by-point responses to the reviewers' comments and modifications included in the revised manuscript.

Reviewer 1: Major comments

- 1. Although authors analyzed the molecular mechanism of NUP-mediated monoallelic expression of *Kcnq1ot1* region in detail using XEN cells, critical question to be solved is whether this mechanism is physiologically important through development. To address this question, knockdown (or knockout by CRISPR) experiment in preimplantation embryo should be performed. RT-qPCR and/or allelic expression analysis of some of core imprinted genes in the blastocysts from siRNA-injected zygote or parthenogenetic embryos would be informative and improve the value of this manuscript significantly.**

We agree with the reviewer that it will be important to analyze NUP-mediated regulation in preimplantation embryos. Thus, we have initiated studies to induce CRISPR deletion in embryos. However, these studies are not completely straightforward and we are developing reagents to analyze gene expression and protein localization *in situ*. This is because we have determined that while NUP-mediated regulation of the *Kcnq1ot1* domain is conserved in extraembryonic endoderm (XEN), embryonic stem (ES) and trophoblast stem (TS) cells, these cells diverge in this regulation. Thus, blastocyst stage embryos, possessing the three stem cell lineages, will require analyses at the blastomere-level. We have shared our results of ES and TS cells in a supplemental file so that the reviewer can see that NUP-mediated imprinted domain regulation is physiologically important in development. Briefly, we found that NUP107, NUP62 and NUP153 regulated paternal *Kcnq1ot1* ncRNA expression and paternal *Kcnq1ot1* domain positioning at the nuclear periphery in ES and TS cells. While NUP107, NUP62 and NUP153 regulated paternal allelic silencing of specific imprinted genes in all three stem cell lines, the genes regulated differed. NUP107, NUP62 and NUP153 were bound at the *Kcnq1ot1* ICR on the paternal allele in ES, TS and XEN cells, as well as at paternal alleles of imprinted genes that had been reactivated upon *Nup* depletion. Compared to XEN cells, where the cohesin complex, but not CTCF, assembled at the paternal *Kcnq1ot1* ICR with NUP107, NUP62 and NUP153, both CTCF and the cohesin complex bound to the paternal *Kcnq1ot1* ICR in a NUP107, NUP62 and NUP153-dependent manner in ES cells, while in TS cells neither CTCF nor cohesin assembled at the paternal *Kcnq1ot1* ICR. These results indicate that NUP107, NUP62 and NUP153 regulate imprinting at the *Kcnq1ot1* domain by a nucleoporin-mediated mechanism in ES, TS and XEN cells, although in a cell lineage-specific manner. Based on reviewers' comments from this manuscript, we are performing

additional analyses in embryonic stem cells and trophoblast stem cells, which will be presented in a separate manuscript.

Minor comments:

1. **P5 line 93: “while SMARCA5 and SMARCA5 regulation is” should be “while SMARCA5 and SMARCA5 and SMARCA5 regulation is”. Related to this title, the conclusion “Smarca5 and Smarcad1 function in a Kcnq1ot1 ncRNA/transcription-dependent manner at the paternal Kcnq1ot1 domain” (P7 line 129-130) is not proved by evidence. What authors found is just 30 to 50% down-regulation of Kcnq1ot1 in Smarca5 and Smarcad1 knockdown cells (Fig. 1b), and this could be coincidence. Without rescue experiments, this conclusion is over-statement.**

We agree with the reviewer that further analysis is required for *Smarca5* and *Smarcad1* depletion. We have removed *Ezh1*, *Smarca5* and *Smarcad1* from the manuscript, and now focus on nucleoporins.

2. **P7 line 4: “The most unexpected candidate is our screen was” seems to be typo. This reviewer thinks it should be “The most unexpected candidate in our screen was”.**

This has been corrected as follows on Page 5, line 91, “One interesting candidate (ranked 7th) from our screen was nucleoporin 107 (*Nup107*) (Table 1).”

3. **P13 line 278” “Supplementary figure 14a” should be “Supplementary figure 14b”**

This has been corrected (Supplementary Fig. 16b).

4. **P13 line 283” “Supplementary figure 14” should be “Supplementary figure 14d”**

This has been corrected (Supplementary Fig. 16d).

5. **Some of sample numbers mentioned in figure legends are strange. For example, in Supplementary figure 19a and b showing IP-Western blotting results, what “n=2-3; n=3 technical replicates’ means? Did authors combine 2 or 3 samples and analyzed together? This sample number information was not shown in Supplementary figure 6. Please clarify them.**

For all IP-Western blot experiments, 2-3 biological samples were analyzed independently, with 3 technical replicates for each sample. We have made changes throughout the text to make this clearer.

6. **Figure 1 legend: Comma and semicolon are confused, “MAT/B6, maternal (red); PAT/CAST; paternal (blue)” should be “MAT/B6, maternal (red); PAT/CAST, paternal (blue)”.**

While Figure 1, containing results from *Smarca5*, *Smarcad1* and *Ezh1*, has been removed from the manuscript, all commas and semicolons have been checked for appropriate usage, and corrected where needed.

7. **Figure 2(f) legend: Explanation for what NP, SP, and NI stand for should be described. Some of commas and semicolons were confused (e.g. “Nup98 siRNA, si153” should be “Nup98 siRNA; si153”).**

Details have been added to new Fig. 2e legend as “The maternal *Kcnq1ot1* domain was randomly positioned within the nucleus (expected **nuclear periphery (NP)** 15%; **sub-nuclear periphery (SP)** 30%; **nuclear interior (NI)** 60%)”. Commas and semicolons have been corrected.

8. **Figure 6 legend: It is unclear what Y axis indicates. Is this % input? How P values are calculated, comparison of total ChIP OR proportion of maternal and paternal ChIP? Also, it is unclear what “Details as described in Figure 2” means, as neither conventional ChIP nor allele-specific ChIP are shown in figure 2.**

The following has been added to Figure 6 legend. “**The Y-axis indicates total and allelic ChIP enrichment levels represented as percent of input.** Allelic proportions are represented as a percent of the total **ChIP** enrichment level. *, significance $p < 0.05$ **of paternal/maternal levels** compared to the siNT **paternal/maternal** control. We have deleted “Details as described in Figure 2” . In Figure 6, the Y axis has also been modified to “Total and allelic enrichment levels” .

9. **Supplementary Figure 2 (B): No explanation about the experimental methods. Although it is assumable this is the result of RT-PCR, authors should specify it clearly.**

We have now added “**NeoR expression was detected by RT-PCR**” to the figure legend. For clarity, we also have added the sample type above the sample name, differentiating between WT and *Cdkn1c*^{+/ Δ neoR} samples.

Reviewer #2:

1. **The screen that is described is not shown to respond to imprinting. No evidence is provided that the selection target (Neo) is regulated in a parental allele-specific manner in XEN cells. Whereas it is possible that regulators of the selected locus contribute to imprinted gene regulation, it must be clearly stated that the results of this assay do not reflect on the roles of the tested shRNA targets in imprinted gene regulation.**

The reviewer is correct. While we show that NeoR is silenced in XENs cells upon paternal transmission, we have not assessed NeoR expression upon maternal transmission. We no longer have these mice to examine this. We have made the necessary changes in text to reflect the reviewers comment. Page 4, lines 75-77, we have added “reactivation of the silent *Cdkn1c*^{*ΔneoR*} allele following depletion would allow for survival and selection of colonies in the presence of neomycin, and thus, identification of epigenetic factors crucial in maintaining its silent state”. In Supplemental Figure 2, we have added, “Note that maternal transmission of *Cdkn1c*^{*ΔneoR*} in XEN, ES and TS cells was not examined to assess whether it remained active when maternally inherited, similar to midgestation tissues (Caspary et al., 1998).

- 2. The specificity of shRNA depletion of the targeted proteins is not clearly established. Whereas the levels of the proteins are shown to be reduced by the corresponding shRNAs, the effects of the shRNAs on other proteins known to influence imprinting are not tested. This deficiency is aggravated by the fact that the effects of a single shRNA directed against most proteins are characterized (though several shRNAs were used against some proteins n the screen). Additionally, no attempt is made to restore protein expression or to use independent approaches (e.g, genetic mutations) to corroborate the results. This deficiency is particularly serious in the case of the depletion of individual subunits of protein complexes whose disruption could influence the levels of the other subunits.**

To address the concern that proteins known to regulate imprinting may be affected by nucleoporin depletion, we examined protein levels for KMT2a, EHMT, EZH2 (Supplementary Figure 22) and CTCF, SMC1 and SMC2 (Supplementary Figure 24a). No changes were found in nuclear levels of these proteins in control and *Nup*-depleted cells.

To address the concern that depleting one nucleoporin may disrupt other subunits in the complex, we measured the levels of various nucleoporins, which are part of the cytoplasmic filaments (NUP358), the nuclear and cytoplasmic rings (NUP93, NUP160, AHCTF1), and the nuclear basket (NUP50 and TPR). No changes were observed in nucleoporin levels of subunits in the nuclear pore complex upon *Nup107*⁻, *Nup62*⁻, *Nup98/96*⁻ and *Nup153*-depletion (Supplementary Figure 17).

- 3. In several experiments, the levels of transcripts are shown as % maternal and % paternal (e.g, Fig. 1, Fig. 2b, Fig. 5). These data are interpreted to show reactivation of the paternal allele, and a role for the protein targeted by shRNA in allele-specific silencing. However, the data do not demonstrate a selective effect on transcription of one allele. The absolute levels of transcripts from each allele should be shown. This would obviate the need to show transcript levels and allele-specific expression separately (e.g. Fig. 1b and 1c).**

We thank the reviewer for this important point. To date, studies on imprinted expression have not had the capacity to measure absolute levels of transcripts from each allele. Instead for relative transcript abundance, allele-specific expression and relative expression compared to a reference tissue have been measured. To determine absolute levels of paternal and maternal expression independently, we have developed highly sensitive, precision assays for analyzing absolute allele-specific transcript copy numbers for 9 genes in the *Kcnq1ot1* domain. This method employs automated droplet digital PCR assays using FAM and HEX strain-specific probes (Supplementary Fig 5). Importantly, it surpasses other conventional approaches. We have left Fig. 1a and 1b on *Kcnq1ot1* to corroborate the droplet digital assay in new Fig. 1c, and have moved previous Figure 5 of imprinted genes transcript levels to Supplementary Figure 12 as a reference. These assays demonstrate that nucleoporins act on the paternal domain exclusively in *Nup107*- and *Nup62*-depleted cells, reducing paternal *Kcnq1ot1* ncRNA copies (Fig. 1), and increasing paternal copies for a subset of protein coding genes (Fig. 5). In *Nup153*-depleted cells, while paternal *Kcnq1ot1* ncRNA copies are reduced and maternal copies are increased (Fig. 1), there was increased paternal copies for a subset of protein coding genes, with no significant change in maternal transcript copy number (Fig. 5).

- 4. The conclusion that Ezh1 function versus Smarca1/5 function does not or does depend on Kcnq1ot1 transcription is not justified. The fact that Kcnq1ot1 bulk transcript level does not or does change has no bearing on whether it is not or is required for function.**

We agree with the reviewer that further analysis is required for *Ezh1*, *Smarca5* and *Smarcad1*. In this manuscript, we now focus on nucleoporins.

- 5. The interpretation that different Nup gene products regulate only the paternal versus maternal Kcnq1ot1 allele is not justified (Fig. 2a, 2b). It is equally plausible that they function at both alleles, but that other proteins confer paternal-specific expression.**

The absolute transcript abundance measured by droplet digital PCR, nucleoporin ChIP, and cohesion ChIP analysis, with corresponding changes in KMT2A, EHMT2 and EZH2 and histone modifications demonstrate that the function of NUP107 and NUP62 are specific to the paternal allele. NUP153 appears to function on both the maternal and paternal *Kcnq1ot1* alleles.

- 6. The differences in Kcnq1ot1 RNA “volume” appear to correlate with transcript levels (Fig. 2). Is there any significance to these differences?**

There was a possibility that there could be a change in transcript abundance, without a change in volume if not all transcripts produced localize at the domain. To examine this relationship, we performed Pearson Correlation Analysis. We have added to Supplementary Fig. 7 legend, “**We also performed a Pearson Correlation Analysis**”

on *Kcnq1ot1* ncRNA expression levels and *Kcnq1ot1* ncRNA cloud volume, and found a strong positive correlation ($r=0.981$), indicating that most *Kcnq1ot1* ncRNAs localize to the *Kcnq1ot1* domain.”

- 7. The bar graphs that plot paternal *Kcnq1ot1* locus locations (Fig. 2f) do not specify the % of cells where the locus was not detected. Differences in detection of the locus in different cell populations can produce the appearance of a change in localization. The probes that were used to determine locus and transcript volumes are not clearly described.**

To the methods, we have added details (lines 553-556) on how FISH signal volumes and distance were measured, “For DNA and RNA FISH, signal volume was measured in Volocity (PerkinElmer) with z-stacks intensity-based thresholding (μm^3), while distances of DNA FISH signal centroids from the nuclear rim were calculated to determine domain positioning (μm) (ImageJ).” In Figure Legend 2, we have added (lines 756-757), “For these analyses, cells with no RNA but detectable DNA FISH signals were included, while those lacking DNA signals were excluded”. To Supplementary Figure 7 we have added, “Nuclei with no DNA FISH signal were excluded (Vehicle, 5%; siNT, 7%; si107, 7%; si62, 5%; si98/96, 6%; si153, 5% of cells). Finally, to Supplementary Figure 8, we have added “Nuclei with no RNA and DNA FISH signal were excluded (Vehicle, 9%; siNT, 11%; si107, 9%; si62, 11%; si98/96, 12%; si153, 8% of cells).”

- 8. The preferential detection of the paternal allele in Nup immunoprecipitates is potentially significant. It is not clear how the data were normalized to account for potential differences in recovery of the paternal and maternal alleles in the ChIP assay. The same concern pertains to the data in Fig. 6 since the recovery of the maternal versus paternal alleles could change in response to siRNA expression and/or transcription of the locus. Input chromatin is not a valid control since recovery in ChIP assays is affected by many factors. The efficiencies of allelic ChIP must be compared with proteins that bind specifically to both alleles with equal efficiency. The cause versus consequence relationship of the histone modifications and transcription is also unclear. If H3K4me3 and H3K9me2 levels were important in determination of the levels of imprinted gene transcription, one would expect that depletion of the corresponding methyltransferases or demethylases would influence the transcript levels.**

The issue of differential allelic recovery in ChIP is an important point. To address this concern, we performed ChIP using histone 3 (H3) pan-antibody, which is expected to be enriched on both parental alleles, at all tested sites, followed by allelic determination. The results showed equal enrichment of both parental alleles at control genes (Supplementary Fig. 9) and regions across the *Kcnq1ot1* domain (Supplementary Fig. 21). In addition, equal enrichment of parental alleles from Input DNA rules out PCR bias (Supplementary Fig. 20).

To partly address the second point, we have performed ChIP analysis using antibodies for KMT2a (methyltransferase that trimethylates H3K4), EHMT2 (methyltransferase that catalyzes dimethylation of H3K9), and EZH2 (methyltransferase that catalyzes methyl groups to H3K27) (Figure 7). We observed a significant decrease/increase in KMT2a, EHMT2 and EZH2 enrichment that corresponded to alterations in histone modification at imprinted genes in the domain upon *Nup107/Nup62* and *Nup153* depletion.

- 9. The in vitro binding data lack important controls to determine if Nups bind DNA specifically. No control DNA fragment is used in the EMSA experiment. Since the specific binding sites are not identified and the characteristics of the probes that are used as negative controls in the DNA affinity precipitation experiment, the data are not convincing.**

To address this important concern on lack of controls, we performed EMSA on nucleoporin-positive (*Vim*, *Shank2*) and -negative (*Dhcr7*) promoter sites using mAb414, NUP107, NUP98 and NUP153 antibodies (Supplementary Figure 9). We observed supershifts at the positive but not at the negative site (Supplementary Figure 10).

- 10. The reported interactions between Nups with RNAPII, CTCF, SMC1 and SMC3 are not well established. The IgG control precipitation may not reflect the full range of non-specific and indirect targets that are precipitated by the α -Nup and anti-RNAPII antibodies that are used. A better control is the precipitation of complexes from cells where the Nup that is recognized by the antibody is depleted.**

The reviewer raises an important point. To address whether interaction between NUPs, CTCF, SMC1 and SMC3 are specific, we performed immunoprecipitation assays in control and *Nup*-depleted cell lysates. More specifically, we performed mAb414 IP in control and *Nup107/62*-depleted cells, and NUP153 IP in control and *Nup153*-depleted cells, followed by Western using CTCF, SMC1, and SMC3 antibodies. In control cells, we observed positive interactions, while in *Nup107/62*-depleted or *Nup153*-depleted cells CTCF, SMC1, and SMC3 recovery from the IP was reduced, suggesting that the interaction is specific (Supplementary Figure 24b).

We further determined whether interaction with CTCF, SMC1A, and SMC3 occurred selectively through NUP107/62, and/or NUP153 by performing IP with mAb414 in *Nup153*-depleted cells and NUP153 IP in *Nup107/62*-depleted cells. NUP107/62 was required for NUP153 and CTCF/SMC1/SMC3 interactions, but NUP153 was not required for mAb414 (NUP107/NUP62) interactions (Fig. 9a). To further delineate this, NUP153 IP was performed with *Nup107* or *Nup62*-depleted lysates. Notably, NUP107 but not NUP62 was required for NUP153 and CTCF/SMC1/SMC3 interactions (Fig. 9b).

- 11. The effects of Nup depletion on SMC1 and SMC3 ChIP efficiencies in the IC3/IC4 regions are the most significant findings relating to their potential roles in ICR regulation. To determine the specificity of these effects, critical controls must be performed to establish if SMC1 or SMC3 binding at other loci where they bind specifically are affected by Nup depletion. Determination of the levels of SMC1 and SMC3 proteins does not establish if their overall levels of chromatin binding are altered. There is also a larger question whether the mechanisms that are investigated pertain to imprinting or if they regulate the genes that are investigated in an allele-independent manner.**

To determine the specificity of SMC1 and SMC3 ChIP in control and nucleoporin-depleted cells, we tested SMC1 and SMC3 binding by ChIP at the *Vim* (NUP153 positive), *Orai2* (NUP153, mAb414 and NUP98 positive), *Shank2* (NUP153, mAb414 and NUP98 positive), and *Dhcr7* (NUP153, mAb414, NUP98 negative) promoter sites in control and nucleoporin-depleted cells. No change in SMC1 and SMC3 recruitment was observed at these sites (Supplemental Figure 25), indicating that binding was nucleoporin-independent. This contrasted with SMC1 and SMC3 binding at the *Kcnq1ot1* domain ICR, which was NUP107, NUP62 and NUP153-dependent (Fig. 8).

Reviewer #3:

- 1. The authors have use XEN cells to identify factors involved in controlling *Kcnq1ot1* imprinted domain but it is not clear if the role of the genes identified is the same in other cell types, can the authors investigate the relevance of their findings in other cell types?**

Please see reply to Reviewer 1's Comment 1.

- 2. Similarly, to further validate a potential role of the nucleoporins in establishing imprinting they should look at other imprinted domains, this would establish either a general role or specific role in imprinting.**

To examine the role for nucleoporins in regulating other imprinted domains, we will develop droplet digital PCR assays with strain-specific FAM and HEX probes, and will perform ChIP sequencing with NUP107 and NUP153 antibodies in control and nucleoporin depleted XEN, ES and TS cells. This analysis is ongoing and will be presented in a separate manuscript.

- 3. The first results section describing validation of EZH1, SMARCA5 and SMARCA5 feels incomplete, the authors investigate a potential role in DNA methylation at the ICR site but do not pursue other possibilities. Perhaps the authors could also investigate their role on histone modifications (particularly applicable to EZH1), it is not clear why this was done for the nucleoporins and not for this set of candidates. Related to this the authors discuss SMARCA5,**

along with CTCF and the cohesin complex, positively regulating transcription of H19. Given this, it is not clear why the authors would not test the hypothesis that SMARAC5 regulates *Kcnq1ot1* transcription in a similar manner.

We agree with the reviewer that further analysis is required for *Ezh1*, *Smarca5* and *Smarcad1*. In this manuscript, we now focus on nucleoporins.

- 4. The authors observe a decrease in *Kcnq1ot1* expression upon knockdown of *Nup107*, *Nup62* and *Nup153*. *Kcnq1ot1* has been shown to recruit EZH2 and this is believed to be the mechanism by which it establishes imprinting. This could be the potential mechanism through which the nucleoporins regulate allelic silencing at certain imprinted genes. The authors should investigate whether EZH2 (PRC2) recruitment is reduced – which would in turn provide a mechanism for the reduced H3K27 levels observed. The role of *Kcnq1ot1* in recruiting should also be included in the discussion.**

Please see reply to Reviewer 2's Comment 8. In the discussion, we have added on Page 20, lines 437-443, "*Interactions between the *Kcnq1ot1* ncRNA and the polycomb repressive complex 2 and EHMT2 could in turn direct repressive epigenetic marks, H3K27me3 and H3K9me2, respectively, to these paternally silent genes. This is support by our data since alterations in KMT2A, EHMT2 and EZH2 binding occurred concomitantly with changes in H3K4me3, H3K9me2 and H3K27me3 levels, respectively, at the *Kcnq1ot1* ICR and imprinted gene promoters upon *Nup107/62* and/or *Nup153* depletion."*

- 5. A number of reports have established the role of b-catenin and Wnt signalling in the function of nucleoporins and *Kcnq1ot1* transcription. The author should explore this link in their cell types and following point 1, in other cell types too.**

Thank you for this suggestion. To explore this potential linkage further, we examined TCF1 consensus sequences to which β -catenin binds, as well as β -catenin occupancy within the mouse *Kcnq1ot1* domain. In the human, β -catenin was found to bind a TCF1 site in the KCNQ1OT1 promoter, regulating the levels of KCNQ1OT1 transcription (Sunamura et al., Sci Rep 6, 2016). While we observed a number of TCF1 consensus motifs in the mouse, the TCF1 site identified in human KCNQ1OT1 promoter was not conserved. We also examine data produced from ChIP-seq using flag-tagged or biotin-tagged β -catenin in ES cells. There was no significant enrichment of β -catenin in the mouse *Kcnq1ot1* promoter, or in the whole *Kcnq1ot1* gene. In fact, there is only one significant site of β -catenin enrichment in the entire domain. In future studies, we can explore this in more detail.

- 6. *Kcnq1ot1* has also been shown to help establish silencing at *Kcnq1* through mediating intrachromosomal interactions As *Kcnq1* is one of the core genes impacted upon, is this interaction lost?**

The impact of nucleoporin depletion on potential intrachromosomal interactions between *Kcnq1ot1* and other genes within the domain is a good question. To examine the role for nucleoporins in regulating other imprinted domains, we will perform HiChIP sequencing with NUP107 and NUP153 antibodies in control and nucleoporin depleted XEN, ES and TS cells. This analysis is ongoing and will be presented in a separate manuscript.

- 7. The material and methods could be improved. There is no description of the method used for methylation analysis. For gene expression the $2\Delta\Delta\text{ct}$ method is used not the $\Delta\Delta\text{ct}$ method. It is also implied the authors used a one-tailed T-test for all analysis. For example a one-sided test assumes the authors have made a hypothesis of the direction of gene expression – NUP62, NUP98 and NUP153 were not included in the original screen and so the ‘expected’ direction of expression changes for genes cannot be known. Therefore a two-tailed test would be more appropriate. I think certain details are missing from the ChIP method – How long was crosslinking carried out for? How much chromatin was used?**

The method for the DNA methylation analysis is in the supplementary material since the figure for this analysis is supplemental. The “ $\Delta\Delta\text{ct}$ method” has been corrected to “ $2\Delta\Delta\text{ct}$ method”. All data have now been analyzed by two-tailed test. Details missing from the ChIP method are now included Page 25, lines 562-566, “Approximately 40,000 cells were used per ChIP reaction. Briefly, cells were cross-linked in 1% formaldehyde for 6 minutes, lysed in SDS buffer and then sonicated (Diagenode or Covaris M220) to shear chromatin into 100-400 bp fragments, followed by immunoprecipitation. Following ChIP, DNA extraction was performed using Chelex beads (BioRad) or ChIP DNA Clean & Concentrator™ (Zymo Research).”

Minor comments

- 1. The writing style – there appears to be a lack of narrative where statements are made but not elaborated upon. For example results Lines 98/99. There are a few examples of where the authors have incorrectly used cohesion complex instead of cohesin complex.**

This has been corrected. “Cohesion” has been changed to “cohesin”.

- 2. Abstract line 28 depletion repeated**

This has been corrected.

- 3. Abstract line 35/36 – the role of nucleoporins is only established for the *Kcnq1ot1* imprinted domains in this manuscript – this sentence implies a general mechanism for all imprinted domains.**

This has been amended to (line 33) the “*Kcnq1ot1* imprinted domain”.

4. Introduction line 43 should be domains not domain

This has been corrected (line 40).

5. Introduction lines 63-65 some protein symbols are missing – in the abstract protein symbols are not included.

This has been corrected (lines 58-60).

6. Results line 93 SMARCA5 is repeated, should be replaced by SMARCAD1.

This has been deleted.

7. Results line 134 why is the most unexpected candidate?

This has been amended to (lines 91-94), “**One interesting** candidate (ranked 7th) **from** our screen was nucleoporin 107 (*Nup107*) (Table 1). **Nucleoporins are proteins that compose nuclear pore complexes that stud the nuclear membrane, allowing nuclear-cytoplasmic transport. A more recent role identified for nucleoporins is gene regulation, although the mechanistic underpinnings of this regulation are not fully understood.**”

8. Results line 138/139 I assume the siRNA with the best (or possibly most consistent) knockdown was chosen?

Two siRNAs were tested individually for consistent RNA depletion, and then together for RNA and protein depletions. To achieve maximum depletion, we employed both siRNAs for each experiment.

9. Results line 338-340 – Kcnq1ot1 has been shown to regulate H3K27me3 thorough EZH2 recruitment. Therefore the alterations of the histone modifications at imprinted gene promoters could be as an indirect result of this.

Please see response to Comment 4, above.

10. Discussion there is no mention of the implication of altered cohesin complex interaction.

The following has been added to the discussion (Page 20, lines 445-462) “**Another explanation is** that NUP107, NUP62 and NUP153 confer **chromatin structure** at the nuclear periphery. **NUP107, NUP62 and NUP153 interactions with the cohesin complex at the nuclear periphery on both side of the *Kcnq1ot1* promoter may enable open chromatin configuration at the paternal *Kcnq1ot1* ICR, enabling *Kcnq1ot1***

transcription. This is supported by reactivation of the maternal *Kcnq1ot1* ncRNA allele coincident with a shift in maternal *Kcnq1ot1* domain positioning at the nuclear periphery upon *Nup153* depletion. Demarcating of active chromatin by NUP107, NUP62, NUP153 and cohesin may separate active chromatin at the paternal *Kcnq1ot1* ICR from repressive chromatin at the paternal alleles of the core group of imprinted genes, blocking their expression. NUP107/NUP62 also bound at the *Osbp15* promoter region, and NUP153 bound at the *Cd81* promoter possibly demarcating repressive chromatin at *Osbp15* and *Cd81* from active chromatin at neighbouring non-imprinted genes in XEN cells. Upon *Nup107/Nup62* and *Nup153* depletion, cohesin binding was reduced at the paternal *Kcnq1ot1* ICR, possibly altering chromatin states at the paternal *Kcnq1ot1* ICR and the core group of imprinted genes, leading to alteration in paternal allelic expression. Furthermore, chromatin structure change at the paternal *Osbp15* and *Cd81* promoters may lead them to adopt the active chromatin structure of neighbouring non-imprinted genes. Thus, further investigations are required to determine the role of NUP107, NUP62 and NUP153 in regulating higher-order chromatin structure at the *Kcnq1ot1* imprinted domain.”

11. Discussion there is no mention of the role of *Kcnq1ot1* ncRNA in regulating PRC2 recruitment to the imprinted domain. Or the role of *Kcnq1ot1* ncRNA in chromatin structure at this locus. There is also evidence for recruitment of G9a – given that H3K9me is also altered this warrants discussion.

Please see response to Comment 4, above.

12. Supplementary figure 11 abstract negative not negatives

This has been corrected to “negative”.

Reviewers' comments:

Reviewer #1 (Remarks to the Author):

From the previously submitted version, the authors removed Smrca part and added some important data to focus on Nup function in the expression regulation of Kcnq1ot1 locus. At the first round, this reviewer asked to analyze Nup-deficient embryo to test whether Nup-mediated imprinting maintenance machinery is commonly used in vivo. Authors have not added such data because molecular mechanism of Nup-mediated imprinting regulation seems to be varied among cell types with proving the results from Nup-deficient ESC and TSC. This reviewer understands that whole story will be too long and complicated if that newly provided data will be integrated into current manuscript. Although physiological importance of Nup-regulated imprinting regulation in vivo is still unclear, authors improved the quality of manuscript by adding data including the digital PCR analysis to measure the absolute number of transcripts and some important control experiments. Therefore, this reviewer think this manuscript is worth considering to be published in Nature communications.

Minor comments

Figure 1: Delocalized X-axis label "si153" of Figure 1b should be corrected.

Figure 3: Since important information was duplicated between Figure 3 and 4, Figure 3 should be moved to supplementary materials.

Reviewer #2 (Remarks to the Author):

The revised manuscript addresses some of the criticisms raised in the original review. The following comments refer to the items of the original review and should be read in the context of the original criticisms.

1. The text falsely claims that the experiment addresses imprinting. The revision does not adequately respond to the criticism.
2. The experiments address a miniscule subset of potentially important regulators. The readers should be alerted to the likelihood of off-target effects and the lack of controls to test this.
3. The independent measurement of allele-specific transcripts is a significant improvement. The figures are difficult to decipher because the colored bars are often invisible. Alternative ways to display/label the data should be considered.
4. The change in focus is appropriate and requires rigorous demonstration of Nup functional roles.
5. The allele-specificity is consistent with the new data. The reasons for the differences in allele specificities in different Nups are unclear and not adequately discussed.
6. The relationship between transcript levels and FISH signal size implies that no active mechanism re-localized the transcripts. If true, this should be stated.
7. The added information corroborates the interpretation.
8. The allele-specific binding of Nups is a key issue. It must be clearly stated if the H3 control was performed in parallel using the same cell extracts that were used to perform the Nup ChIP. The roles of histone modifications and the (indirect?) regulation of histone methyltransferase binding by Nups remain a source of uncertainty and some confusion. The effects of Nup depletion on methyltransferase occupancy do not appear to be allele-specific (a few exceptions), whereas the changes in histone modifications are (in more cases) allele-preferential. This needs to be noted and discussed. The lack of information concerning the cause-effect relationship with transcription should be recognized.
9. It should be clearly stated if the experiments using different DNA probes were conducted in parallel using the identical extracts. It would also be appropriate to show images of the entire gel, including the wells.
10. The epitope-depletion experiment supports the interpretation and addresses the criticism.
11. It must be clearly stated if the analysis of SMC1A and SMC3 binding at the control loci was

performed using the same ChIP samples that were used to analyze binding at the IC3 and IC4 regions. If show, one of the control loci should be included in the main figure to demonstrate the locus-specific effect of Nup depletion.

Reviewer #3 (Remarks to the Author):

In the response to the reviewers, the authors attempted to answer to all of the major and minor queries and performed additional interesting experiments. However, as detailed below, the responses fell short in testing the general role of the discovered Nup proteins in genomic imprinting.

The authors have stated that certain experiments are ongoing and will be represented in a separate manuscript. I assume that this is because the scope of the manuscript has been altered and so the experiments are no longer relevant to the narrative of the study. However, I believe it is the responsibility of the authors to justify not including these experiments (i.e. a paragraph explaining why they have not been included in the present study), and unfortunately this is not the case. I feel there is no justification for not investigating the impact upon other imprinted regions, it's an obvious question, and in my opinion leaves the study feeling incomplete. The phrase "genomic imprinting" is very frequently used in the abstract and background information, therefore exploring the impact of nucleoporins on other imprinted genes is essential. That being said, the experiments that have been done have been performed to a high standard, and there is no doubt that work done is of interest to the field. The authors may want to restrict the interpretation by referring to the mechanism of control of the *Kcnq1ot1* locus without generalising it as a mechanism to control genomic imprinting (e.g. Starting the abstract with "Genomic imprinting", title containing "imprinting", "Line 42 "In this study, we employed the *Kcnq1ot1* imprinted domain as a model" etc).

Regarding the question on the role of nucleoporins in regulating of *Kcnq1ot1* locus in other cell types (a similar question asked by Reviewer1 point 1), it is not clear to me how ES and TS cells are the right alternative cell types when the Supplementary figure2b shows that the *NeoR* gene is being expressed in ES and TS cells, compared to XEN cells, where it is silenced (which was the basis of this screen). In addition, why is this separate supplementary information? The relevance of the nucleoporins in the control of the *Kcnq1ot1* locus in other cell types will be a question asked by many readers if the manuscript is published, I assume this work wasn't done solely to satisfy the reviewers' question.

In response to reviewer 2 point 6 it is unclear to me how could the authors apply the Pearson Correlation analysis for the correlation between the *Kcnq1ot1* expression levels and ncRNA cloud volume when, as far as I understand, it is not possible to get these two parameters from the same cell. In addition, the statistical analysis is only mentioned in a supplementary figure legend but I didn't find it explained anywhere in the manuscript. They would need to justify precisely why and how they used Pearsons Correlation as a test because as far as this reviewers is concerned, it is an inappropriate test.

Finally, the manuscript needs to be significantly shortened. I appreciate that there has been a large amount of work done but for the benefit of the reader and the work itself, this volume of data became difficult to follow efficiently.

We thank the reviewers for their time to review our modified manuscript and for their insightful comments. We have addressed their concerns and feel that these revisions have improved the clarity of the manuscript. Below are point-by-point responses to the reviewers' comments and modifications included in the revised manuscript.

Reviewers' comments:

Reviewer #1 (Remarks to the Author):

1. Figure 1: Delocalized X-axis label “si153” of Figure 1b should be corrected.

Delocalized X-axis label 'si153' has been corrected.

2. Figure 3: Since important information was duplicated between Figure 3 and 4, Figure 3 should be moved to supplementary materials.

We agree with the reviewer. Figure 3 has now been moved to the supplemental figures (Supplementary Figure 10).

Reviewer #2 (Remarks to the Author):

1. The text falsely claims that the experiment addresses imprinting. The revision does not adequately respond to the criticism.

Throughout the manuscript, we now refer specifically to the *Kcnq1ot1* domain, rather than to genomic imprinting in general. To further clarify the RNA interference screen experiment, we have changed the language as follows. Page 4, lines 69-71, “**Multiple epigenetic factors silence a paternal *NeoR* cassette** To identify epigenetic factors involved in **paternally-inherited *NeoR* silencing, as a proxy for paternal allelic silencing of imprinted genes in the *Kcnq1ot1* domain ...**”.

2. The experiments address a miniscule subset of potentially important regulators. The readers should be alerted to the likelihood of off-target effects and the lack of controls to test this.

We agree with the reviewer that off-target effects may occur with the shRNA screen. However, there is greater confidence in the candidates where multiple independent colonies and/or multiple hairpins were recovered. We have added the following changes to the text. Page 5, lines 85-86, “In total, 41 epigenetic modifiers were identified (Table 1), **with stronger candidates having multiple independent colonies and/or multiple hairpins recovered.**” We have also added on Page 5, lines 89-92, “**Given the chance of off-target effects, validation of candidates will be required to delineate their role in paternal allelic silencing at the *Kcnq1ot1* domain. Here, we focused on the candidate nucleoporin 107 (*Nup107*) (Table 1).**”

3. The independent measurement of allele-specific transcripts is a significant improvement. The figures are difficult to decipher because the colored bars are often invisible. Alternative ways to display/label the data should be considered.

We have now increased the size of the box plots. This will enable the reader to visually differentiate maternal and paternal bars more easily.

5. The allele-specificity is consistent with the new data. The reasons for the differences in allele specificities in different Nups are unclear and not adequately discussed.

This is a good question. For example, why did NUP107 and NUP62 bind the paternal *Kcnq1ot1* ICR, while NUP153 binds to both the maternal and paternal *Kcnq1ot1* ICR. We have added a section in the discussion to expand upon this. Page 20, lines 428-434, following the sentences on KMT2A, EHMT2 and EZH2, “Recently, NUP153 was identified to play a role in PRC1 silencing of developmental genes in ES cells²⁷. This is consistent with reactivation of the normally silent maternal *Kcnq1ot1* allele upon *Nup153* depletion. Thus, nucleoporins may interact with different macromolecular epigenetic complexes to regulate gene expression. For example, NUP107, NUP62 and NUP153 may interact with a complex containing KMT2A to enable *Kcnq1ot1* ncRNA expression, while NUP153 may engage with a repressive complex to regulate maternal *Kcnq1ot1* ncRNA silencing.”

6. The relationship between transcript levels and FISH signal size implies that no active mechanism re-localized the transcripts. If true, this should be stated.

In keeping with reviewer 3, comment 1, we have removed the statistical method to correlate transcripts levels and FISH signal. In keeping with this, we refrain from adding any additional statement.

8. The allele-specific binding of Nups is a key issue. It must be clearly stated if the H3 control was performed in parallel using the same cell extracts that were used to perform the Nup ChIP. The roles of histone modifications and the (indirect?) regulation of histone methyltransferase binding by Nups remain a source of uncertainty and some confusion. The effects of Nup depletion on methyltransferase occupancy do not appear to be allele-specific (a few exceptions), whereas the changes in histone modifications are (in more cases) allele-preferential. This needs to be noted and discussed. The lack of information concerning the cause-effect relationship with transcription should be recognized.

9. It should be clearly stated if the experiments using different DNA probes were conducted in parallel using the identical extracts. It would also be appropriate to show images of the entire gel, including the wells.

11. It must be clearly stated if the analysis of SMC1A and SMC3 binding at the control loci was performed using the same ChIP samples that were used to analyze binding at the IC3 and IC4 regions. If show, one of the control loci should be included in the main figure to demonstrate the locus-specific effect of Nup depletion.

Respond 8, 9 and 11: All analyses were performed on the same cell lines from 3-4 biological samples. While different experiments were performed using the same cell extracts from these lines, there was insufficient material to conduct all experiments on these extracts. Thus, multiple cell extracts from the same cell lines were used after confirming *Kcnq1ot1* ncRNA expression levels (maintained in controls, decreased in *Nup107-*, *Nup62-* and *Nup153-*depleted, and increased in *Nup98-*depleted extracts), allelic *Cdkn1c* expression (maternal-specific for controls, paternal reactivated in *Nup107-*, *Nup6-2* and *Nup153-*depleted extracts), and mRNA depletion in *Nup* siRNA-treated extracts.

Reviewer #3 (Remarks to the Author):

1. As detailed below, the responses fell short in testing the general role of the discovered Nup proteins in genomic imprinting. I believe it is the responsibility of the authors to justify not including these experiments (i.e. a paragraph explaining why they have not been included in the present study), and unfortunately this is not the case. I feel there is no justification for not investigating the impact upon other imprinted regions, it's an obvious question, and in my opinion leaves the study feeling incomplete. The phrase "genomic imprinting" is very frequently used in the abstract and background information, therefore exploring the impact of nucleoporins on other imprinted genes is essential. The authors may want to restrict the interpretation by referring to the mechanism of control of the *Kcnq1ot1* locus without generalising it as a mechanism to control genomic imprinting (e.g. Starting the abstract with "Genomic imprinting", title containing "imprinting", "Line 42 "In this study, we employed the *Kcnq1ot1* imprinted domain as a model" etc). The relevance of the nucleoporins in the control of the *Kcnq1ot1* locus in other cell types will be a question asked by many readers if the manuscript is published, I assume this work wasn't done solely to satisfy the reviewers' question.

We now included a paragraph that described the focus of the current study on nucleoporins. Page 5, Line 85-92, "In total, 41 epigenetic modifiers were identified (Table 1), **with stronger candidates having multiple independent colonies and multiple hairpins recovered**. Candidates included factors (RNF2, EZH2, EED, SUV420H1 and DNMT1) previously shown to play a role in *Kcnq1ot1* imprinted domain regulation^{3,7,9,13-16,20}, validating our screening strategy. However, they were recovered at a lower frequency than other candidates. Given **the chance of off-target effects, validation of candidates will be required to delineate their role in paternal allelic silencing at the *Kcnq1ot1* domain. Here, we will focus on** the candidate nucleoporin 107 (*Nup107*)

(Table 1).” We also added a final paragraph to the discussion. Page 21, lines 458-464, “Overall, **dynamics of nucleoporin-mediated regulation of the *Kcnq1ot1* domain highlights the complexity** of epigenetic processes required to maintain imprinting **across a large** imprinted domain **during early development. Further investigations are now required to determine whether this mechanism is conserved in embryonic and trophoblast stem cells, as well as at other imprinted domains.** Finally, the identification of **additional candidate** epigenetic factors opens a new dimension **of investigations into understanding** imprinted domain regulation.” Finally, throughout the manuscript, we now refer specifically to the *Kcnq1ot1* domain, rather than to genomic imprinting in general.

2. In response to reviewer 2 point 6 it is unclear to me how could the authors apply the Pearson Correlation analysis for the correlation between the *Kcnq1ot1* expression levels and ncRNA cloud volume when, as far as I understand, it is not possible to get these two parameters from the same cell. In addition, the statistical analysis is only mentioned in a supplementary figure legend but I didn't find it explained anywhere in the manuscript. They would need to justify precisely why and how they used Pearsons Correlation as a test because as far as this reviewers is concerned, it is an inappropriate test.

We thank the reviewer for pointing this out. Upon further research and analysis, we agree this is not possible. We have removed this statistical analysis from supplemental figures.

3. Finally, the manuscript needs to be significantly shortened. I appreciate that there has been a large amount of work done but for the benefit of the reader and the work itself, this volume of data became difficult to follow efficiently.

The main body of the text in the manuscript was and currently under 5000 words, which falls within the guidelines for Nature Communications. We have modified the text in some sections to add further clarification and ease of reading.

Reviewers' comments:

Reviewer #2 (Remarks to the Author):

The re-revised manuscript addresses most of the criticisms raised in the original reviews. The main remaining concerns relate to inadequate design or description of controls for ChIP experiments

8, 11. Several experiments show reduced efficiencies of ChIP in response to Nup depletion by particular antibodies, at specific genes, and in some cases different alleles of the same gene. In each of these cases, it is essential to demonstrate that some other antibody produced different relative efficiencies of precipitation using the same chromatin preparations. This comparison should be made for the same combination of genes or alleles in a parallel assay. The efficiencies of ChIP can vary between different cell populations for reasons unrelated to differences in binding/occupancy. The figure legends need to state which specific ChIP assays were performed using the same chromatin preparation. In all cases where different chromatin preparations were used to perform assays that are presented in the same figure, this needs to be explicitly stated. Many of the controls are now presented in supplementary figures. Any experiments where the controls are presented in supplementary data must state if the same extracts were used.

9. The same applies to the in vitro binding assays. Comparisons between samples must explicitly state whether or not the experiments were performed in parallel using the same cell extracts.

Reviewer #3 (Remarks to the Author):

The authors have revised the manuscript and acknowledge the lack of evidence for a generalised role of the nucleoporins identified in genomic imprinting, outside the locus analysed, and in different cell types. This changes the original narrative so it remains at the discretion of the editors to decide if this study merits publication in Nature Communications. Nevertheless, the screening results may be a valuable resource for the genomic imprinting community.

Regarding the response to the Pearsons Correlation question - this further raises some concerns over other potentially flawed statistical analyses in the paper. I am not a statistician, I can provide only limited feedback.

I agree with Reviewer 2 regarding providing full gel images (which as far as I can see the authors haven't done but I might be wrong) as Fig 8b left panel CTCF and SMC1A bands are too similar with CTCF looking a bit more overexposed.

Reviewers' comments:

Reviewer #2:

8, 11. Several experiments show reduced efficiencies of ChIP in response to Nup depletion by particular antibodies, at specific genes, and in some cases different alleles of the same gene. In each of these cases, it is essential to demonstrate that some other antibody produced different relative efficiencies of precipitation using the same chromatin preparations. This comparison should be made for the same combination of genes or alleles in a parallel assay. The efficiencies of ChIP can vary between different cell populations for reasons unrelated to differences in binding/occupancy. The figure legends need to state which specific ChIP assays were performed using the same chromatin preparation. In all cases where different chromatin preparations were used to perform assays that are presented in the same figure, this needs to be explicitly stated. Many of the controls are now presented in supplementary figures. Any experiments where the controls are presented in supplementary data must state if the same extracts were used.

8. H3 control ChIP was performed with the same extracts as the main body figures. We have added to the text “Note: the same extracts used in Supplementary Figure 22 were used as Figure 3”.

11. For ChIP at other genes outside of the *Kcnq1ot1* imprinted domain, the same nuclear extracts were used in one experiment. The three other replicates which used different nuclear extracts demonstrated the same results. We have also added “Note: Different extracts were used in Supplementary Figure 26 compared to Supplementary Figure 9, and Figure 3”.

9. The same applies to the in vitro binding assays. Comparisons between samples must explicitly state whether or not the experiments were performed in parallel using the same cell extracts.

The EMSA was performed with different extracts. Here, the assay is not quantitative, but instead aims to determine whether there was a supershift or not. This is why the gel wells are not commonly shown for this assay. Furthermore, in this case, where the supershifts are located near the top of the blot, exposure from the wells bleeds over to obscure the supershift bands. We have added to the text “Note: the two extracts in (a) were two of the same extracts used in (b)”.

Reviewer #3:

The authors have revised the manuscript and acknowledge the lack of evidence for a generalised role of the nucleoporins identified in genomic imprinting, outside the locus analysed, and in different cell types. This changes the original narrative so it remains at the discretion of the editors to decide if this study merits publication in Nature Communications. Nevertheless, the screening results may be a valuable resource for the genomic imprinting community.

We can understand Reviewer 3's sentiment. There are papers that have examined other imprinted domains simply by assessing one imprinted gene per domain or DNA methylation at ICRs.

However, unless ChIP-seq or RNA-seq has been performed, papers recently published in the field (Nature Medicine, Science, Cell Reports) do not investigate other domains. For these papers, the lack of evidence for a generalized role of specific proteins of interest in genomic imprinting does not diminish the findings or importance of the paper. Furthermore, intriguing findings of our manuscript are that there was not a domain-wide effect nor a change in imprinted methylation upon nucleoporin depletion. This means that other imprinted domains would need to be investigated in totality. Testing only the commonly investigated imprinted genes in a domain for allelic expression changes as a diagnostic would be insufficient. Instead, the same degree of in depth investigation as this manuscript would be required at other domains.

We have added the following to the manuscript “**The question also remains whether nucleoporins act as global imprinting regulators, operate at a subset of imprinted domains, or are specific to the *Kcnq1ot1* imprinted domain. As a preliminary query, we analyzed NUP153 DAM ID-seq from ES cells, and found enrichment peaks at multiple ICRs, including *Gtl2*, *Snrpn*, *Igf2r*, *Peg1* and *Peg10*, in addition to the at *Kcnq1ot1* ICR (Jacinto et. al., 2015). This suggest a more generalized role for NUP153 and other nucleoporins in imprinted domain regulation.**”

Finally, given the fact that besides the polycomb response complex and histone modifiers, there has been little advancement in identifying other molecules that regulate imprinted domains. Thus, this study represents a significant advancement in the field. Furthermore, we are laying the groundwork for future advances, ultimately connecting the dots between chromatin structure, nuclear architecture and imprinted domain regulation.

Regarding the response to the Pearsons Correlation question - this further raises some concerns over other potentially flawed statistical analyses in the paper. I am not a statistician, I can provide only limited feedback.

The Pearsons Correlation has been removed from the Supplementary file.

I agree with Reviewer 2 regarding providing full gel images (which as far as I can see the authors haven't done but I might be wrong) as Fig 8b left panel CTCF and SMC1A bands are too similar with CTCF looking a bit more overexposed.

Representative gel images of antibody validation are now provided in the Supplementary Figures 3 and 24. Regarding Fig. 8b, the same Western blot is probed subsequently with 3 different antibodies. The bands for CTCF and SMC1A are at different locations on the Western blot.